# Immunogenicity to COVID-19 mRNA vaccine third dose in people living with HIV

Alessandra Vergori [1,11] ✉, Alessandro Cozzi Lepri[2,11], Stefania Cicalini[1], Giulia Matusali[3], Veronica Bordoni[4], Simone Lanini[1], Silvia Meschi[3], Roberta Iannazzo[1], Valentina Mazzotta [1], Francesca Colavita[3], Ilaria Mastrorosa[1], Eleonora Cimini[4], Davide Mariotti [4], Lydia De Pascale[1], Alessandra Marani[5], Paola Gallì[5], AnnaRosa Garbuglia[3], Concetta Castilletti [3], Vincenzo Puro[6], Chiara Agrati[4], Enrico Girardi[7], Francesco Vaia[5], Andrea Antinori[1] & HIV-VAC study group*

In order to investigate safety and immunogenicity of SARS-CoV-2 vaccine third dose in people living with HIV (PLWH), we analyze anti-RBD, micro-neutralization assay and IFN-γ production in 216 PLWH on ART with advanced disease (CD4 count <200 cell/mm$^3$ and/or previous AIDS) receiving the third dose of a mRNA vaccine (BNT162b2 or mRNA-1273) after a median of 142 days from the second dose. Median age is 54 years, median CD4 nadir 45 cell/mm$^3$ (20–122), 93% HIV-RNA < 50 c/mL. In 68% of PLWH at least one side-effect, generally mild, is recorded. Humoral response after the third dose was strong and higher than that achieved with the second dose (>2 log$_2$ difference), especially when a heterologous combination with mRNA-1273 as third shot is used. In contrast, cell-mediated immunity remain stable. Our data support usefulness of third dose in PLWH currently receiving suppressive ART who presented with severe immune dysregulation.

The Severe Acute Respiratory Syndrome Coronavirus 2 (SARS-CoV-2) pandemic has led to more than 270 million confirmed cases of COVID-19 and about 5 million deaths[1], as reported by the World Health organization up to December, 27th 2021. Effective vaccines licensed against SARS-CoV-2, with more than 7 million doses administered worldwide[1] have proved to be a highly successful strategy to reduce the disease burden, particularly in people at high risk of developing a severe COVID-19[2]. In addition, real-world data suggest that an effective vaccination is successful in preventing severe disease and death even in the presence of variants of concern (VoCs)[3,4] such as Gamma and

Delta variants, and many countries regulatory authorities indicate a third additional dose (AD) as necessary to maintain an adequate protection to SARS-CoV-2 and to the VoCs, especially for susceptible sub-populations[5,6].

More recently, serious concerns are arising for the emergence of the B.1.1.529 (Omicron) variant because it has been shown that vaccine effectiveness is significantly lower than that seen with the delta variant, limiting the antibody-mediated neutralization and possibly increase the risk of reinfections[7–10] Several studies recently supported a significant increase of neutralization against Omicron after a booster

[1]HIV/AIDS Unit, National Institute for Infectious Diseases Lazzaro Spallanzani IRCCS, Rome, Italy. [2]Centre for Clinical Research, Epidemiology, Modelling and Evaluation (CREME), Institute for Global Health, UCL, London, UK. [3]Laboratory of Virology, National Institute for Infectious Diseases Lazzaro Spallanzani IRCCS, Rome, Italy. [4]Laboratory of Cellular Immunology and Clinical Pharmacology, National Institute for Infectious Diseases Lazzaro Spallanzani IRCCS, Roma, Italy. [5]Health Direction, National Institute for Infectious Diseases Lazzaro Spallanzani IRCCS, Roma, Italy. [6]Risk Management Unit, National Institute for Infectious Diseases Lazzaro Spallanzani IRCCS, Roma, Italy. [7]Scientific Direction, National Institute for Infectious Diseases Lazzaro Spallanzani IRCCS, Roma, Italy. [11]These authors contributed equally: Alessandra Vergori, Alessandro Cozzi Lepri. *A list of authors and their affiliations appears at the end of the paper. ✉e-mail: alessandra.vergori@inmi.it

dose, although this increase was lower than that observed with ancestral type or Delta[7–11], suggesting a cross-reactivity of neutralizing antibody responses[10,11].

Although with discrepancies between studies, people living with HIV (PLWH) appear to be a high-risk group for adverse clinical outcomes from COVID-19, with some evidence for higher hospitalization and mortality rates[12–14] that might be due to a poor neutralizing antibody titres reflecting a reduced antibody response to SARS-CoV-2 natural infection[15] as well as to the presence of additional comorbidities and low socioeconomic status or occupational risk[16].These data are consistent with the observation that HIV infection may favor a poor serological response to other viral agents, such as influenza[17].

Up to date, only one randomized, phase 2/3 trial in UK have described both immunogenicity to a prime-boost dosing of the chimpanzee adenovirus-vectored ChAdOx1-nCoV-19 vaccine in PLWH on stable ART and with CD4 counts > 350 cells/mm$^3$[18] and its durability after 6 months[19]. This follow-up analysis showed a decline in humoral and cell-mediated immunity, at 6 months, although with no significant difference compared to a cohort of HIV uninfected individuals vaccinated with the same vaccine.

Regarding mRNA vaccines in PLWH, to date only few observational studies exist and all showed a satisfactory humoral[20–26] and T cell immune response[20] in PLWH on ART and with CD4 T cell counts above 200 cell/mm$^3$ after primary vaccination cycle, but no information was available about efficacy and safety of additional or booster dose in HIV-infected population, especially in people with low CD4 count.

We previously reported that mRNA vaccination is able to elicit a robust humoral and cellular immune response against SARS-CoV-2 in most of PLWH receiving ART, particularly in those with full immune recovery after suppressive therapy, even though such response was significantly poorer in PLWH with current CD4 T-cell < 200/mm$^3$ compared to those with > 500 cell/mm$^3$ and HIV-uninfected controls. These results suggest that chronic persistent dysregulation in ART-treated population may affect the effector immune response to SARS CoV2 vaccination[27].

On September 10th, 2021, the administration of a third dose of anti-SARS-CoV-2 mRNA vaccine was approved in Italy to be given after >28 days after completion of the primary vaccination cycle in PLWH (to be intended as a full additional dose vaccine) who presented with a CD4 T cell count <200/mm$^3$ and/or previous AIDS at the time of their first dose. The aims of this analysis were to investigate reactogenicity and degree of immunogenicity after the third dose in PLWH, to compare the levels to those achieved after the second dose and to evaluate the association between immune response and current CD4 T cells count and specific vaccine sequence.

## Results

A total of 216 PLWH were included in the analysis (PCDR = 44; ICDR = 96; HCDR = 76). The main characteristics of HIV-infected participants according to current CD4 T cell count strata at time of third dose are reported in Table 1. Briefly, the median age was 54 years old (IQR 47, 59); all HIV patients were on ART at time of third dose, 92.6% had HIV-RNA < 50 copies/mL with a median time since HIV diagnosis of 7 years (3–12) and of 5 years (2–8) since AIDS, if diagnosed; 6.9% with a diagnosis of cancer; the three groups significantly differed for CD4 count nadir ($p = 0.007$), previous AIDS diagnosis ($p < 0.001$), time since AIDS diagnosis ($p = 0.001$), HIV-RNA ($p < 0.001$) at third dose. Proportion of PLWH with HIV-RNA lower than 50 copies/mL at the time of third dose was 79.5% in PCDR, 93.8% in ICDR, and 98.7% in HCDR ($p < 0.001$). More details on CD4 transition in CD4 count from baseline (the time of first dose) to $T_0$ (time of third dose) are shown in supplementary table 1. The median time from the date of primary vaccination series to third dose was 142 days (132–156), longer for PCDR than ICDR and HCDR [154 (134,159) versus 145 (132,157) and 138 (130–151); $p = 0.006$].

The HCWs group was mainly composed by female subjects [72/98 (73.5%) vs 39/216 PLWH (18%); $p < 0.0001$], with a younger median age 44 years old (IQR 32–52) than PLWH (54 years old (47–59), the median time between the third dose and response measurement was 16 days (14–18), shorter than that observed in PLWH [33 days (30–35)], as well as, the interval of response measurement between $T_{-1}$ and $T_0$ was was significantly longer than PLWH [285 days (280, 291) vs. 117 (103,126)].

We identified $n = 44/214$ (20%) participants who received BNT162b2 vaccination for all the 3 doses and $n = 80/214$ (37%) to mRNA-1273 for all the three doses, $n = 57/214$ (26.6%) received as primary series BNT162b2 and for third dose mRNA-1273, whereas, $n = 31/214$ (14.4%) received mRNA-1273 as primary vaccination and BNT162b2 as third dose. There were two participants with a sequence involving not mRNA vaccines (one ChAdOx-ncov-19 followed by mRNA-1273 and one Ad26.COV2.S followed by mRNA-1273) which have been excluded from the regression analysis.

In a subset of the study population for whom a value of the response was available at time $T_{-1}$, we compared the level of immunogenic response post third dose with that achieved 1 month after the 2nd dose of vaccination. The humoral response elicited by third dose was on average stronger than the titres elicited by the primary vaccination 1 month after the completion of 2 doses vaccination cycle (time $T_{-1}$) although different timepoints post-vaccination are compared.

The comparison of the mean log$_2$ of anti-RBD IgG between $T_{-1}$ and $T_1$ was performed in $N = 169/216$ patients and these means were: 9.8 BAU/mL (SD 2.9) at $T_{-1}$ and 11.8 (SD 2.1) at $T_1$ ($p = 0.003$) with an estimated mean log$_2$ increase of 2.0 (SD 1.6) ($p < 0.0001$) (Fig. 1A).

The mean log$_2$ of nAbs (available for $N = 75/216$ patients) were: 4.9 (SD 2.2) at $T_{-1}$ and 8.3 (SD 2.4) at $T_1$ ($p = 0.018$) with a mean increase of 3.4 (SD 2.1) ($p < 0.0001$) (Fig. 1B).

The mean log$_2$ of IFN-γ (available for $N = 74/216$) were 6.8 pg/mL (SD 3.2) at $T_{-1}$ and 7.2 (SD 2.9) at $T_1$ ($p = 0.39$) with a non-significant mean increase [0.4 pg/mL (SD 2.4; $p = 0.12$) (Fig. 1C).

We found an association between the level of CD4 count a $T_0$ and the observed variations in humoral response from the peak after the 2nd dose ($T_{-1}$) and 15 days after 3D ($T_1$) (Fisher test $p = 0.003$). The difference was driven by the contrast between ICDR vs. HCDR (mean log$_2$ 0.5, std error 0.14, $p = 0.002$) (Supplementary Fig. 2A–C); in contrast, for the other two responses there was little evidence for an association with CD4 count: nAbs (fisher test $p = 0.06$) (Supplementary Fig. 3A–C) and for IFN-γ ($p = 0.23$) (Supplementary Fig. 4A–C).

### Overall response to the third dose

The $T_1$ values for anti-RBD were truncated for 30/216 (14%) at the cut off of 11,360 BAU/mL. The overall proportion of $T_1$ value above the (log$_2$ transformed) cut off of 7.1 BAU/mL was 214/216 (99%); three out of 4 (75%) participants in whom their anti-RBD level was <7.1 BAU/mL at $T_{-1}$ showed a level above this cut-off post third dose.

In terms of the quantitative response in PLWH, the unadjusted anti-RBD IgG mean log$_2$ were 7.3 BAU/mL (SD 2.8) at $T_0$ and 11.8 (SD 2.2) at $T_1$, (Fig. 2A). There was evidence for a significant increase from $T_0$ to $T_1$ (4.5 mean log$_2$ (SD 1.9) [paired t-test ($p < 0.0001$)].

In HCWs, the unadjusted anti-RBD IgG mean log$_2$ were 6.1 BAU/mL (SD 1.1) at $T_0$ and 12.0 (SD 0.9) at $T_1$, (Fig. 3A). There was evidence for a significant increase from $T_0$ to $T_1$ (5.9 mean log$_2$ (SD 1.2) [paired t-test ($p < 0.0001$)].

If compared PLWH with the control group of HCWs and after adjusting for gender, age, and time difference in the exact day interval between the booster dose and the measured responses (between $T_0$ and $T_1$ and between $T_{-1}$ and $T_0$) we did not observe a difference of anti RBD at $T_1$ between PLWH and HCWs [1.27 (−0.72, −3.27); $p = 0.212$] (Table 2).

The $T_1$ values for nAbs were truncated for 107/216 (50%) at the cut off of 1:1280. The proportion of $T_1$ value above the (log$_2$ transformed) cut off of 1:10 was 206/216 (96%). Fourteen out of 20 (70%) participants

**Table 1 | Characteristics of the study population**

| Characteristics | Current CD4 count (cells/mm³) | | | *p*-value[*] | Total |
|---|---|---|---|---|---|
| | HCDR 500 + | ICDR 201-500 | PCDR 0-200 | | |
| | N = 76 | N = 96 | N = 44 | | N = 216 |
| Age, years | | | | 0.074 | |
| Median (IQR) | 52 (47, 58) | 55 (47, 60) | 57 (48, 63) | | 54 (47, 59) |
| Female, *n*(%) | 12 (15.8) | 17 (17.7) | 10 (22.7) | 0.632 | 39 (18.1) |
| Caucasian, *n*(%) | 63 (82.9) | 63 (65.6) | 30 (68.2) | 0.035 | 156 (72.2) |
| Nadir CD4 count, cells/mm³ | | | | 0.007 | |
| Median (IQR) | 83 (26, 168) | 41 (16, 92) | 40 (15, 76) | | 45 (20, 122) |
| Time from HIV diagnosis, years | | | | 0.135 | |
| Median (IQR) | 6 (4, 11) | 6 (3, 11) | 15 (2, 25) | | 7 (3, 12) |
| Time from AIDS diagnosis, years | | | | 0.001 | |
| Median (IQR) | 6 (4, 9) | 4 (2, 8) | 2 (2, 5) | | 5 (2, 8) |
| AIDS, *n*(%) | 73 (96.1) | 79 (82.3) | 15 (34.1) | <0.001 | 167 (77.3) |
| Year of starting ART | | | | 0.668 | |
| Median (IQR) | 2015 (2011, 2017) | 2015 (2011, 2019) | 2014 (2000, 2020) | | 2015 (2010, 2018) |
| VL < = 50 at T0, *n*(%) | 65 (95.6) | 80 (87.9) | 23 (56.1) | <0.001 | 168 (84.0) |
| VL < = 50 at T1, *n*(%) | 74 (98.7) | 90 (93.8) | 35 (79.5) | <0.001 | 199 (92.6) |
| Time from dose to response, days | | | | 0.668 | |
| Median (IQR) | 16 (15, 18) | 16 (14, 19) | 16 (15, 20) | | 16 (14, 18) |
| Time from second dose to third dose, days | | | | 0.006 | |
| Median (IQR) | 138 (130, 151) | 145 (132, 157) | 154 (134, 159) | | 142 (132, 156) |
| BMI | | | | 0.532 | |
| Median (IQR) | 67 (27, 80) | 51 (24, 76) | 65 (59, 71) | | 65 (27, 75) |
| Cancer, *n*(%) | 9 (11.8) | 5 (5.2) | 1 (2.3) | 0.094 | 15 (6.9) |
| Autoimmune disease, *n*(%) | 1 (1.3) | 0 (0.0) | 0 (0.0) | 0.401 | 1 (0.5) |
| Cardiopathy, *n*(%) | 1 (1.3) | 0 (0.0) | 0 (0.0) | 0.401 | 1 (0.5) |
| CKD, *n*(%) | 4 (5.3) | 5 (5.3) | 8 (18.6) | 0.017 | 17 (8.0) |
| COPD, *n*(%) | 2 (2.7) | 4 (4.3) | 1 (2.3) | 0.783 | 7 (3.3) |
| MI, *n*(%) | 0 (0.0) | 1 (1.1) | 0 (0.0) | 0.534 | 1 (0.5) |
| Hypertension, *n*(%) | 3 (4.0) | 11 (11.7) | 7 (16.3) | 0.074 | 21 (9.9) |
| Mild liver disease, *n*(%) | 9 (12.0) | 13 (13.8) | 11 (25.6) | 0.122 | 33 (15.6) |
| Severe liver disease, *n*(%) | 0 (0.0) | 1 (1.1) | 4 (9.3) | 0.003 | 5 (2.4) |
| Corticosteroids therapy, *n*(%) | 0 (0.0) | 0 (0.0) | 0 (0.0) | | 0 (0.0) |
| Immunosuppressive therapy, *n*(%) | 0 (0.0) | 0 (0.0) | 0 (0.0) | | 0 (0.0) |
| No. of comorbidities, | | | | 0.055 | |
| Median (IQR) | 1 (1, 1) | 1 (1, 2) | 1 (1, 2) | | 1 (1, 2) |
| Vaccine sequence, *n*(%) | | | | 0.020 | |
| 3 doses of mRNA-1273 | 36 (47.4) | 37 (38.5) | 8 (19.0) | | 81 (37.5) |
| 2 doses of mRNA-1273 + 3rd dose of BNT162b2 | 14 (18.4) | 14 (14.6) | 3 (7.1) | | 31 (14.3) |
| 2 doses of BNT162b2 + 3rd dose of mRNA-1273 | 16 (21.1) | 25 (26.0) | 17 (40.5) | | 58 (27.1) |
| 3 doses of BNT162b2 | 10 (13.2) | 20 (20.8) | 14 (33.3) | | 44 (20.3) |
| 2 doses of ChAdOx ncov19 + 3rd dose of mRNA-1273 | 1 | 0 | 0 | | 1 (0.4) |
| 1 dose of Ad26.COV2.S + 1 dose of mRNA-1273 | 1 | 0 | 0 | | 1(0.4) |

[*]Chi-square or Kruskal-Wallis test as appropriate, 2-sided no adjustment for multiple comparisons.
*IQR* Interquartile range, *ART* Antiretroviral therapy, *VL* Viral load, *BMI* Body mass index, *CKD* Chronic kidney diseases, *COPD* Chronic obstructive pulmonary diseases, *MI* Myocardial infarction.

in whom their nAbs level was < 1:10 at $T_{-1}$ showed a level above this cut-off post third dose.

Regarding the quantitative response in PLWH, the unadjusted nAbs titres mean $\log_2$ were 4.9 (SD 2.1) at $T_0$ and 8.7 (SD 2.1) at $T_1$, (Fig. 2B). There was evidence for a significant increase from $T_0$ to $T_1$ (3.7 mean $\log_2$ (SD 2.2) [paired *t*-test (*p* < 0.0001)].

In HCWs, the unadjusted nAbs titres mean $\log_2$ were 3.4 (SD 1.4) at $T_0$ and 7.9 (SD 1.3) at $T_1$, (Fig. 3B). There was not a significant increase from $T_0$ to $T_1$ [4.6 mean $\log_2$ (SD 1.7); paired *t*-test (*p* < 0.0001)].

If compared PLWH with the control group of HCWs and after adjusting for gender, age and difference in the exact days interval between the booster dose and the measured responses (between $T_0$ and $T_1$ and between $T_{-1}$ and $T_0$) we did not observe a difference for nAbs at $T_1$ PLWH and HCWs [1.24 (−1.37, 3.85);*p* = 0.353] (Table 2).

The overall proportion of $T_1$ value above the ($\log_2$ transformed) cut off of 12 pg/mL was 186/216 (86%); only 4 out of 11 (36%) participants in whom their IFN-γ level was <12 pg/mL at $T_{-1}$ showed a level above this cut-off post third dose.

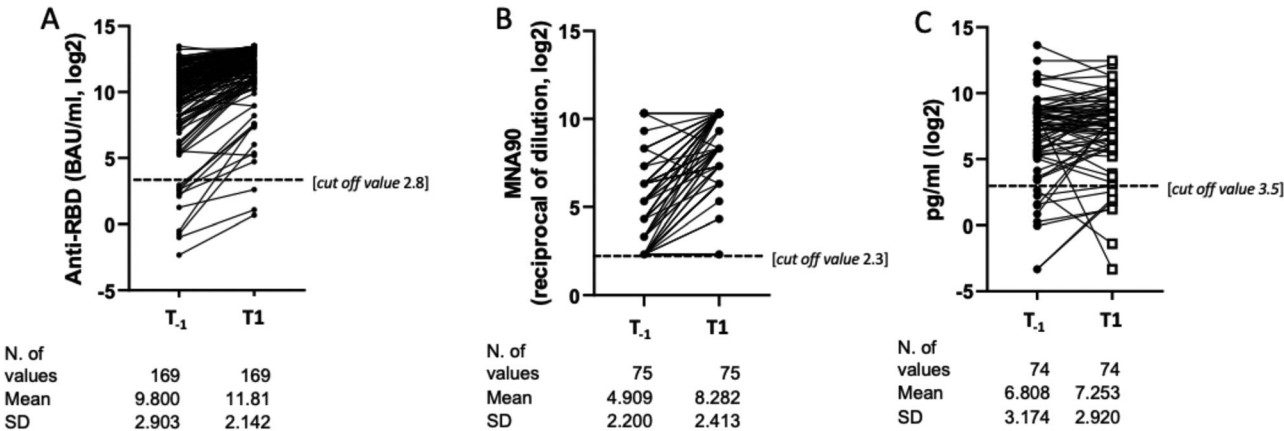

**Fig. 1 | Comparison of the mean log₂ of anti-RBD IgG, MNA₉₀ and IFN-γ from 1 month response after the 2nd dose (T₋₁) and after the 3ʳᵈ dose (T1). A–C** Mean log₂ change of anti-RBD IgG (**A**),MNA₉₀ (**B**) and IFN-γ (**C**) from 1 month response after the 2nd dose (T₋₁) and after the 3ʳᵈ dose (T1) [Paired t-test, 2-sided no adjustment for multiple comparisons]. Mean Log₂ changes of anti-RBD IgG (**A**) and MNA₉₀ (**B**) significantly increased from T₋₁ to T1 ($p < 0.0001$; $p < 0.0001$; respectively but IFN-γ (**C**, $p = 0.39$) Source data are provided as a Source Data file.

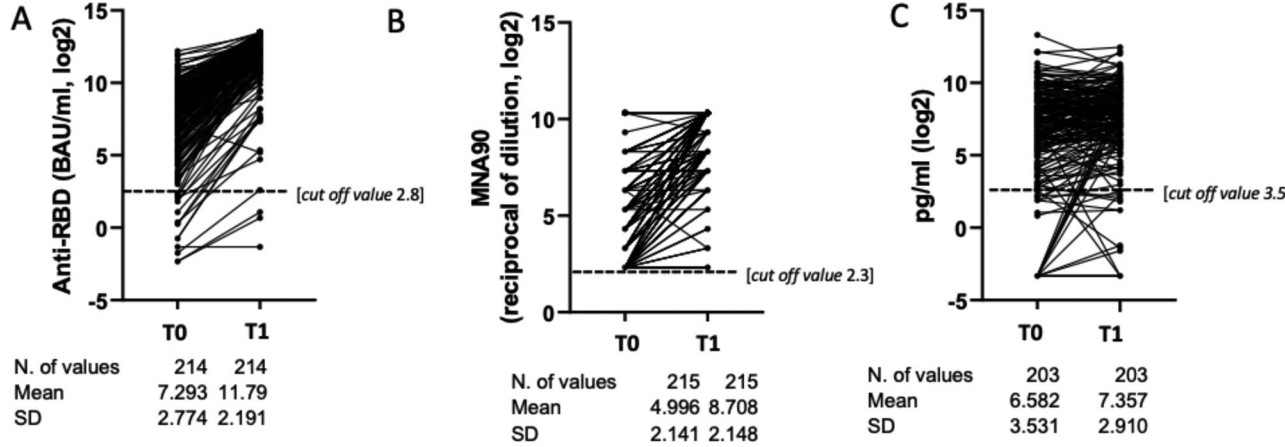

**Fig. 2 | Overall Log₂ change of anti RBD Ig, MNA₉₀ and IFN-γ from T0 to T1; [Paired t-test, 2-sided no adjustment for multiple comparisons]. A** Mean Log₂ values of anti-RBD IgG. **B** MNA₉₀. **C** IFN-γ significantly increased from T0 to T1 ($p < 0.0001$, $p < 0.0001$ and $p = 0.003$, respectively). Source data are provided as a Source Data file.

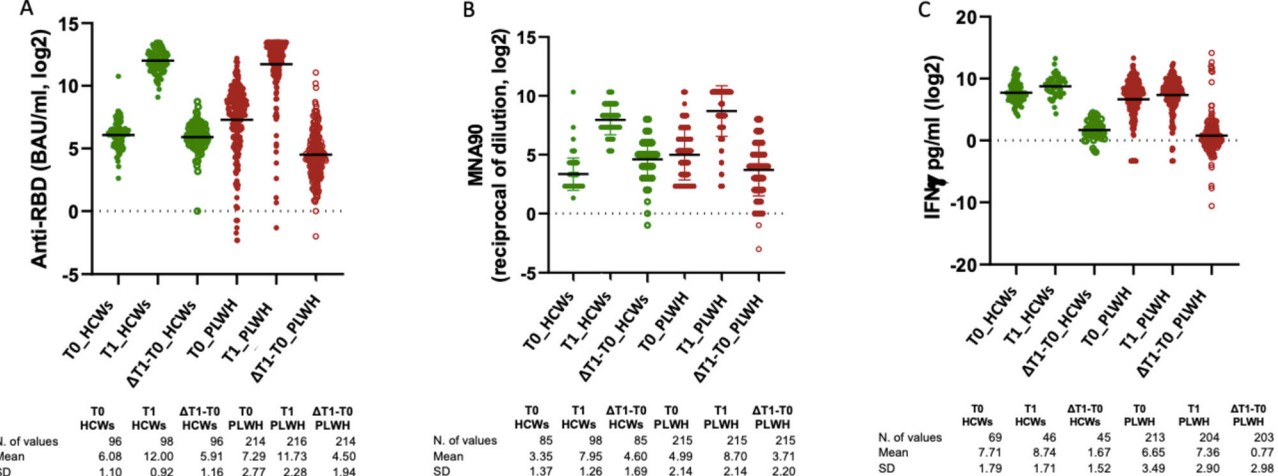

**Fig. 3 | Changes of anti-RBD IgG, MNA₉₀ and IFN-γ from T0 to T1 in HCWs and PLWH.** Log₂ change of anti-RBD IgG (**A**), MNA₉₀ (**B**) and IFN-γ (**C**) from T0 to T1 in HCWs and PLWH; There was evidence for a significant increase of anti-RBD IgG, nAbs and IFN-γ from $T_0$ to $T_1$ ($p < 0.0001$, $p < 0.0001$ and $p = 0.0015$, respectively); ($p$-values are Bonferroni-corrected values). Green circles represent mean values of anti-RBD IgG, MNA₉₀0, and IFN-γ in HCWs, red circles represent mean values of anti RBD IgG, MNA₉₀, and IFN-γ in PLWH. Source data are provided as a Source Data file.

**Table 2 | Adjusted difference in means from fitting an ANCOVA model (contrasts between groups)**

| | Unadjusted and adjusted mean difference at T1 in anti-RBD (log2 scale) HIV vs. HCW | | | |
| --- | --- | --- | --- | --- |
| | Unadjusted Mean (95% CI) | *p*-value | Adjusted* mean (95% CI) | *p*-value |
| RBD-binding IgG | −0.87 (−1.20, −0.53) | <0.001 | 1.27 (−0.72, 3.27) | 0.212 |
| nAb titres | 0.04 (−0.43, 0.52) | 0.853 | 1.24 (−1.37, 3.85) | 0.353 |
| IFN-γ | −1.18 (−1.90, −0.46) | 0.001 | −4.98 (−8.52, −1.44) | 0.006 |

*adjusted for value at T0, age, gender and time difference between T0 and T1 and between T-1 and T0; Z statistic, 2-sided no adjustment for multiple comparisons.

Concerning the quantitative response in PLWH, the unadjusted IFN-γ mean $\log_2$ were 6.6 pg/mL (SD 3.5) at $T_0$ and 7.36 (SD 2.9) at $T_1$ (Fig. 2C). There was evidence for a significant increase from $T_0$ to $T_1$ (0.8 mean $\log_2$ (SD 3.0) [paired t-test ($p = 0.003$)].

In HCWs, the unadjusted IFN-γ mean $\log_2$ were 7.7 pg/mL (SD 1.8) at $T_0$ and 8.7 (SD 1.7) at $T_1$ (Fig. 3C). There was evidence for a significant increase from $T_0$ to $T_1$ (0.7 mean $\log_2$ (SD 3.0) [paired *t*-test ($p = 0.0015$)].

If compared with the control group of HCWs and after adjusting for gender, age and difference in the exact days interval between the booster dose and the measured responses (between T0 and T1 and between T-1 and T0) we observed a significant lower mean difference for IFN-γ at $T_1$ in PLWH [−4.98 (−8.52, −1.44); $p = 0.006$] than HCWs (Table 2).

In terms of the qualitative response, 15 days after the third dose, anti-RBD-binding IgG response (>7.1 BAU/mL) was elicited in 95.5% of PCDR, 100% of ICDR, 100% of HCDR (Fisher exact test $p = 0.04$).

Only 2 participants (both in PCDR) did not show anti-RBD response < 7.1 BAU/mL and because of the small number of events, it was not possible to estimate the odds ratio according to the CD4 count strata.

In contrast, there was no evidence for a difference in the average increase by CD4 count strata by multivariable ANOVA (Fisher test $p = 0.15$) (Fig. 4A–C).

Specifically, the unadjusted anti-RBD IgG mean $\log_2$ were 5.1 BAU/mL (SD 3.7) at $T_0$ and 10 (SD 3.7) at $T_1$ in PDCR, 7.4 BAU/mL (SD 2.4) at $T_0$ and 12.1 (SD 1.5) at $T_1$ in HCDR and 8.4 BAU/mL (SD 1.7) at $T_0$ and 12.3 (SD 1.2) at $T_1$ with no significant differences among groups (Fisher test, $p = 0.19$).

The results from the truncated linear regression analysis carried no evidence for an overall effect of CD4 count on RBD-binding IgG (global Wald test $p = 0.20$) (Table 3).

Mean variations over $T_0$–$T_1$ of anti-RBD IgG levels after the third dose according to CD4 count strata are shown in Supplementary Fig. 5A.

Only one participant with anti-RBD IgG level < 7 BAU/mL was found and he/she received all three doses with BNT162b2 vaccination group.

The mean $\log_2$ increase of anti-RBD IgG comparing $T_0$ to $T_1$ according to vaccine sequence were: 4.5 (SD 2.1) in BNT162b2 group, 4.0 (1.4) in mRNA-1273 group, 5.9 (1.7) in 2BNT162b2 + mRNA-1273 group and 3.3 (1.6) in 2mRNA-1273 + BNT162b2 group (Fisher $p < 0.0001$ in the age-adjusted ANOVA); there was evidence for a difference for four individual contrasts: 2BNT162b2 + mRNA-1273 vs. mRNA-1273 group (mean $\log_2$ difference 0.95, std error 0.15; Bonferroni-corrected $p < 0.0001$), 2BNT162b2 + mRNA-1273 vs. 2mRNA-1273 + BNT162b2 (1.30, std error 0.19; $p < 0.0001$), 2BNT162b2 + mRNA-1273 vs. BNT162b2 (0.69, std error 0.17; $p = 0.004$) and finally 2mRNA-1273 + BNT162b2 vs. BNT162b2 (0.61, std error 0.20; $p = 0.02$) (Supplementary Fig. 6A).

The results from the truncated linear regression analysis confirmed an overall association with vaccine sequence strata (global Wald test $p = 0.0007$) and a significant difference for all four individual contrasts (see below).

In terms of the qualitative response, 15 days after the third dose, nAbs (defined as titres > 1:10) was elicited in 86.3% of PCDR, 97.9% of ICDR, 98.7% of HCDR (Fisher exact $p = 0.001$). In the multivariable logistic regression model, using HCDR as the comparator, PCDR showed a largely increased risk of failing to achieve nAbs > 1:10, after adjusting for the main identified confounders (age, time from HIV diagnosis, CD4 nadir, HIV-RNA < 50 vs > 50 copies/mL at the time of third dose, days from the date of 2nd dose, vaccine sequence and concomitant cancer), although not statistically significant [aOR 5.04 (95% CI 0.22, 115.1); $p = 0.31$] (Table 2). Similarly, there was no evidence for a difference in the average increase by CD4 count strata by multivariable ANOVA (Fisher $p = 0.41$) (Fig. 3D–F). Mean delta changes of nAbs titres after the third dose according to CD4 count strata are shown in Supplementary Fig. 5B. Specifically, the unadjusted nAbs mean $\log_2$ were 3.7 (SD 2.2) at $T_0$ and 7.3 (SD 2.9) at $T_1$ in PDCR, 5.1 (SD 2.1) at $T_0$ and 9.0 (SD 1.9) at $T_1$ in HCDR and 5.7 (SD 1.8) at $T_0$ and 9.1 (SD 1.5) at $T_1$ with no significant differences among groups (Fisher test, $p = 0.18$).

Similarly, the truncated linear regression analysis did not show an overall association both with CD4 count strata and magnitude of nAbs (global Wald $p = 0.26$) (Table 3) and risk of non response to the third dose (Table 4).

Finally, the mean $\log_2$ increase of nAbs comparing $T_0$ to $T_1$ according to vaccine sequence were: 4.09 (SD 2.57) in BNT162b2 group, 3.32 (1.98) in mRNA-1273 group, 4.59 (2.02) in 2BNT162b2 + mRNA-1273 group and 2.77 (1.85) in 2mRNA-1273 + BNT162b2 (age-adjusted ANOVA F $p = 0.0003$); there was evidence for a difference for three contrasts: 2BNT162b2 + mRNA-1273 vs. mRNA-1273 group (mean log difference 0.63, std error 0.18; $p = 0.0006$), HtM vs. HtP (0.91, std error 0.23; $p = 0.0001$), and BNT162b2 group vs. 2mRNA-1273 + BNT162b2 (0.65, std error 0.24; $p = 0.008$) (Supplementary Fig. 6B). The results from the truncated linear regression analysis confirmed an overall association with vaccine sequence strata (global Wald test $p < 0.0001$) and a significant difference for all individual contrasts (Table 5).

In terms of the qualitative response, IFN-γ (defined as levels >12 pg/mL) was elicited in 70% of PCDR, 95.7% of ICDR, 97.2% of HCDR (chi-square $p < 0.0001$).

There was no evidence for a difference in the average increase by CD4 count strata by multivariable ANOVA (Fisher $p = 0.23$) (Table 3 and Fig. 4G–I). Mean delta changes of IFN-γ levels after the third dose according to CD4 count strata are shown in Supplementary Fig. 5C.

Specifically, the unadjusted IFN-γ mean $\log_2$ were 4 pg/mL (SD 3.6) at $T_0$ and 4.7 (SD 3.7) at $T_1$ in PDCR, 7 (SD 3.2) at $T_0$ and 7.7 (SD 2.2) at $T_1$ in HCDR and 7.4 (SD 3.3) at $T_0$ and 8.5 (SD 2.2) at $T_1$ with no significant differences among groups (Fisher test, $p = 0.87$).

In the multivariable logistic regression model using HCDR as the comparator, PCDR showed and increased risk of failing to achieve IFN-γ > 12 pg/mL, after adjusting for the main identified confounders (age, time from HIV diagnosis, CD4 nadir, HIV-RNA at the time of third dose, days from the date of 2nd dose, vaccine sequence and concomitant cancer), although not statistically significant [2.48 (0.29, 21.56); $p = 0.41$] (Table 4).

The mean $\log_2$ increase of IFN- γ comparing $T_0$ to $T_1$ according to vaccine sequence were: 0.47 (SD 3.79) in BNT162b2 group, 0.96 (3.09) in mRNA-1273 group, 0.72 (2.36) in 2BNT162b2 + mRNA-1273 and 0.74

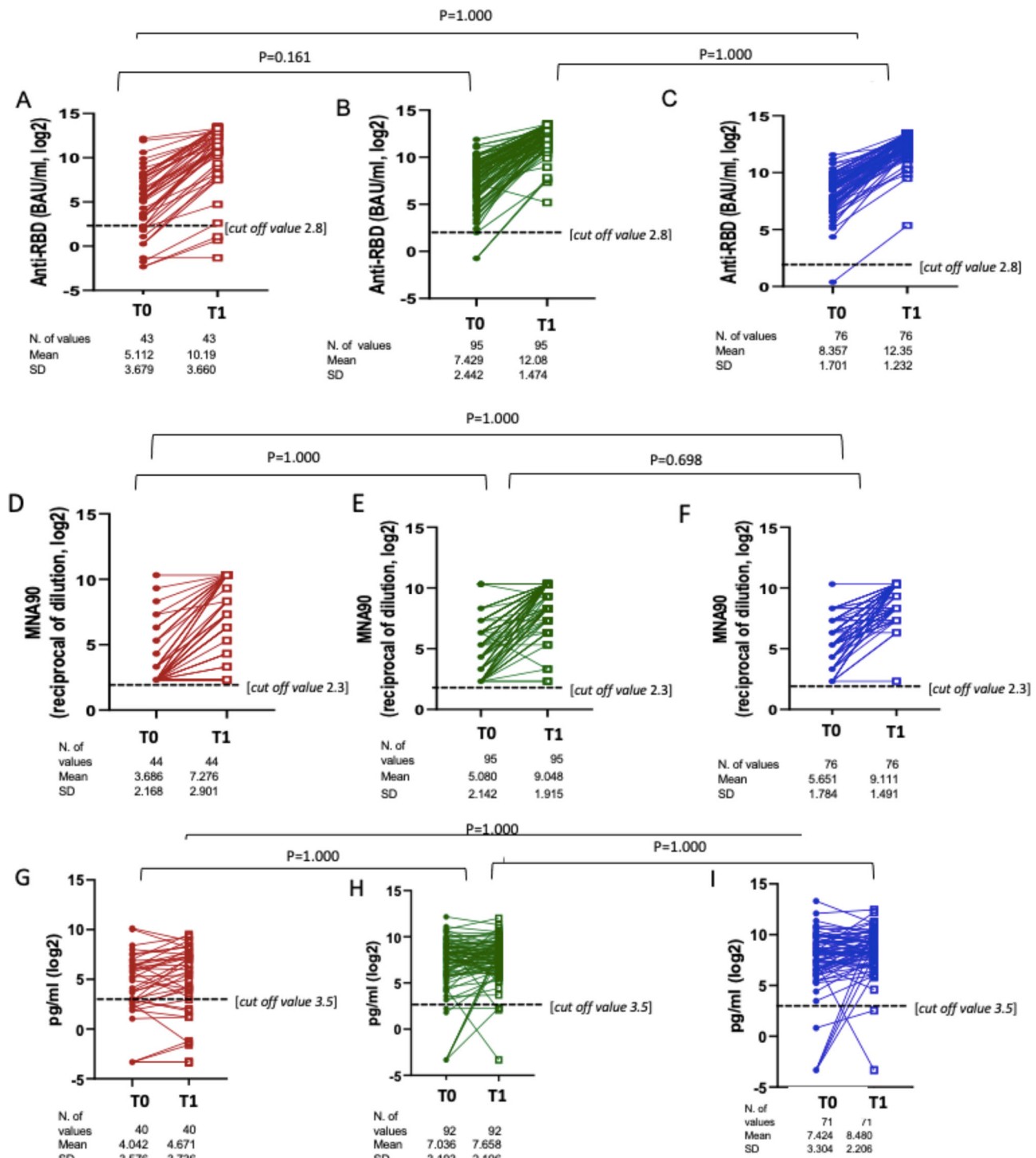

**Fig. 4 | Changes of anti-RBD IgG, MNA₉₀ and IFN-γ from T0 to T1 in PLWH according to immunodeficiency status.** A−C Log₂ change of anti-RBD IgG from T0 to T1 according to immunodeficiency status: PCDR (**A**), ICDR (**B**), HCDR (**C**). **D−F** Log₂ increase of nAbs (MNA₉₀) from T0 to T1 according to immunodeficiency status: PCDR (**D**), ICDR (**E**), HCDR (**F**). **G−I** Log₂ increase of IFN-γ from T0 to T1 according to immunodeficiency status: PCDR (**G**), ICDR (**H**), HCDR (**I**) (ANOVA, 2-sided with Bonferroni adjusted *p*-values for multiple comparisons). Red dots represent mean values of anti RBD IgG, MNA90 and IFN-γ before the third dose, red squares represent mean values of anti RBD IgG, MNA90 and IFN-γ weeks after the third dose in PCDR; green dots represent mean values of anti RBD IgG, MNA90 and IFN-γ before the third dose, green squares represent mean values of anti RBD IgG, MNA₉₀ and IFN-γ 2 weeks after the third dose in ICDR; Blue dots represent mean values of anti RBD IgG, MNA₉₀ and IFN-γ before the third dose, blue squares represent mean values of anti RBD, MNA₉₀ and IFN-γ 2 weeks after the third dose IN HCDR. Source data are provided as a Source Data file.

(2.55) in 2mRNA-1273 + BNT162b2, with no evidence for a difference between the groups (age adjusted *p* = 0.92, Supplementary Fig. 6C; Table 5).

Participants were asked to report side-effects in days 0−7 after the third dose and overall 68% (146/216) of them reported at least one symptom. Among them, 95% reported mild symptoms, 14% moderate symptoms, and 2% severe symptoms not requiring hospitalization. Specifically, 82% complained about inoculation site pain, 44% fatigue, 38% myalgia, 37% fever, 25% headache, 23% shivering and 15% swelling; other symptoms less frequently reported were rash, cough, sore

**Table 3 | Adjusted mean differences from fitting an ANCOVA model (contrasts between groups)**

| | Unadjusted and adjusted mean difference at T1 (log2 scale) | | | |
|---|---|---|---|---|
| | Unadjusted Mean (95% CI) | p-value | Adjusted* mean (95% CI) | p-value |
| **RBD-binding IgG** | | | | |
| HCDR | 0 | | 0 | |
| ICDR | 0.66 (0.09, 1.24) | 0.025 | 0.50 (−0.08, 1.08) | 0.093 |
| PCDR | 1.09 (0.38, 1.80) | 0.003 | 0.05 (−0.84, 0.94) | 0.910 |
| **nAb titres** | | | | |
| HCDR | 0 | | 0 | |
| ICDR | 0.51 (−0.16, 1.17) | 0.136 | 0.27 (−0.47, 1.01) | 0.478 |
| PCDR | 0.13 (−0.69, 0.95) | 0.755 | −0.70 (−1.84, 0.45) | 0.234 |
| **IFN-γ** | | | | |
| HCDR | 0 | | 0 | |
| ICDR | −0.43 (−1.36, 0.49) | 0.358 | −0.03 (−1.15, 1.10) | 0.962 |
| PCDR | −0.43 (−1.59, 0.73) | 0.470 | −0.43 (−2.18, 1.31) | 0.628 |

*adjusted for age, time from HIV diagnosis, CD4 count nadir, VL < = 50 copies/mL at T0, days from the date of 2nd dose, vaccine sequence, and concomitant cancer.
&from the adjusted model; Global Wald test, 2-sided no adjustment for multiple comparisons.

**Table 4 | CD4 count- OR of non-response from fitting a logistic regression analysis**

| | Logistic regression of the risk of no response to third dose | | | | |
|---|---|---|---|---|---|
| | Unadjusted Odds ratio (95% CI) | p-value | Adjusted* Odds ratio (95% CI) | p-value | &Type III p-value |
| **CD4 count (cells/mm³)** | | | | | |
| **nAbs titres** | | | | | |
| HCDR | 1 | | 1 | | 0.410 |
| ICDR | 1.61 (0.14, 18.13) | 0.699 | 2.28 (0.15, 33.93) | 0.550 | |
| PCDR | 11.84 (1.38, 101.9) | 0.024 | 7.45 (0.33, 168.1) | 0.206 | |
| **IFN-γ** | | | | | |
| HCDR | 1 | | 1 | | 0.221 |
| ICDR | 1.59 (0.28, 8.94) | 0.598 | 0.55 (0.07, 4.66) | 0.584 | |
| PCDR | 15.00 (3.15, 71.37) | <0.001 | 2.93 (0.35, 24.47) | 0.320 | |

*adjusted for age, time from HIV diagnosis, CD4 count nadir, VL < = 50 copies/mL at T0 and concomitant cancer.
&from the adjusted model; Global Wald test, 2-sided no adjustment for multiple comparisons.

throat, rhinorrhoea, heavy breathing, nausea, vomit or diarrhea which were overall reported by 30.8% of the participants (Fig. 5). The data carried no evidence for a difference in the prevalence of reported side-effects according to CD4 count strata with the only exception of shivering which was more frequently reported in HCDR (37% vs 19% in ICDR and 13% in PCDR; $p = 0.023$).

The side-effects frequency distribution according to vaccine sequence groups showed that 2BNT162b2 + mRNA-1273 group experienced a higher proportion of symptoms as compared to other combinations (Supplementary Fig. 7).

## Discussion

In this analysis, we found that a third dose of mRNA anti-SARS-CoV-2 vaccine induced a strong humoral and T specific cell response in PLWH eligible for third dose who previously received a complete mRNA 2-dose vaccination cycle. Interestingly, anti RBD IgG, neutralizing antibodies and T cell-mediated response all showed a significant increase. Of note, the level of anti RBD IgG, neutralizing antibodies achieved after the third dose was even higher than that observed one month after the 2nd dose of the primary cycle. Although no significant association could be found with the current level of CD4 count, our data cannot rule out a difference in both the magnitude of response

and risk of no-response when comparing participants with poor CD4 count recovery on ART (PCDR) with those high CD4 count recovery on ART (HCDR). Conversely, the level of T cell mediated response appeared to be more stable comparing values achieved post 2nd dose with those observed post the third dose.

The comparison with a HIV negative control group (HCWs) did not show significantly different anti-RBD and nAbs responses. In contrast, the data carried significant evidence for a lower mean difference in IFN-gamma between PLWH and controls. Thus, these are important results as they show that despite a significant response to a third dose, PLWH appear to have an impaired T-cell mediated response as compared to the general population.

These findings are in line with that recently observed on the characterization of the humoral and SARS-CoV-2 T-cell specific response in PLWH following a mild COVID19, which seem comparable between HIV positive and HIV negative subjects. Moreover, we found a significant difference in term of T-cell specific response in PLWH which results lower than that observed in HIV negative control group and we argue that the overall magnitude of SARS-CoV-2-specific T cell responses relates to the size of the naive CD4 T cell pool, suggesting that inadequate immune reconstitution on ART, could hinder immune responses to SARS-CoV-2 and vaccine effectiveness in PLWH[28].

The observed increase in humoral response in our setting is also consistent with the hypothesis that third dose induces a robust B cell memory response[29], previously elicited by the primary vaccination series and highlights the fact that the SARS-CoV-2 mRNA vaccines are able to stimulate a satisfactory humoral response even in immunocompromised patients such as those with low CD4 count and a previous or current diagnosis of AIDS. These data could be relevant as they add new insights on immune response to supplemental mRNA vaccine in PLWH.

Recently, a matched case-control on humoral response to primary mRNA vaccination cycle in HIV positive and HIV negative showed that PLWH had lower surrogate virus neutralization test response and a trend towards lower IgG response, particularly among those with lower CD4 + T-cell counts and who received the BNT162b2 vaccine (vs mRNA1273), highlighting the need to identify groups that have reduced response to SARS-CoV-2 vaccination in order to set an optimal vaccination schedule with a third additional dose[30].

These response rates are remarkable if compared with those observed in other immunocompromised patients. The rate of anti-RBD response after a third dose in individuals with chronic lymphocytic leukemia (CLL) was of 72% and was even lower when restricting to patients on chemoimmunotherapy (60%)[31]. In non- or low-responders on hemodialysis to standard vaccination, a third dose was able to induce effective antibody titres in only about 70% of patients[32].

However, our findings are comparable with the results of a recent report on immunogenicity in individuals with hematologic and solid cancers[33], and in another study an immediate antibody response to booster administration of the BNT162b2 vaccine was observed in almost all patients with solid organ tumors, including those receiving active systemic chemotherapy[34].

Both the primary vaccination cycle and the third dose were able to increase the Spike-specific T cell response. Nevertheless, differently from the antibody production, the T cell response elicited by the third dose was similar than that achieved by the primary vaccination cycle, suggesting that a fully T-cell immunization is still achieved with the first two doses. The T cell response that contracted overtime after the first two doses can be effectively boosted by the additional third dose. Most of our PLWH with severe immunodeficiency at the time of their first vaccine dose showed an anti-RBD IgG response (95.5%), nAbs response (86.3%) and T cell immunity (70%) after the third dose, and these figures appear to be remarkable in the light of the severe and persistent immunologic dysregulation in this population, due to a reduced T and B cell functionality induced by residual inflammation

**Table 5 | Vax sequence model - OR of non response from fitting a logistic regression analysis**

| | Logistic regression of the risk of no response to additional dose | | | | |
| --- | --- | --- | --- | --- | --- |
| | Unadjusted | | | Adjusted* | |
| | Odds ratio (95% CI) | *p*-value | Odds ratio (95% CI) | *p*-value | &Type III *p*-value |
| **Vaccine sequence** | | | | | |
| | **nAbs titres** | | | | |
| 2BNT162b2 + mRNA-1273 | 1 | | 1 | | 0.326 |
| mRNA-1273 | 0.46 (0.07, 2.82) | 0.398 | 0.45 (0.07, 2.82) | 0.391 | |
| 2mRNA-1273 + BNT162b2 | | | | | |
| BNT162b2 | 1.32 (0.25, 6.86) | 0.744 | 1.01 (0.18, 5.58) | 0.994 | |
| | **IFN-γ** | | | | |
| 2BNT162b2 + mRNA-1273 | 1 | | 1 | | 0.074 |
| mRNA-1273 | 0.41 (0.09, 1.77) | 0.230 | 0.39 (0.09, 1.74) | 0.218 | |
| 2mRNA-1273 + BNT162b2 | 0.36 (0.04, 3.21) | 0.358 | 0.33 (0.04, 3.01) | 0.325 | |
| BNT162b2 | 2.42 (0.73, 8.05) | 0.148 | 1.96 (0.57, 6.81) | 0.288 | |

*age-adjusted.
&from the adjusted model; Unpaired t-test, 2-sided with Bonferroni adjusted p-values for multiple comparisons.

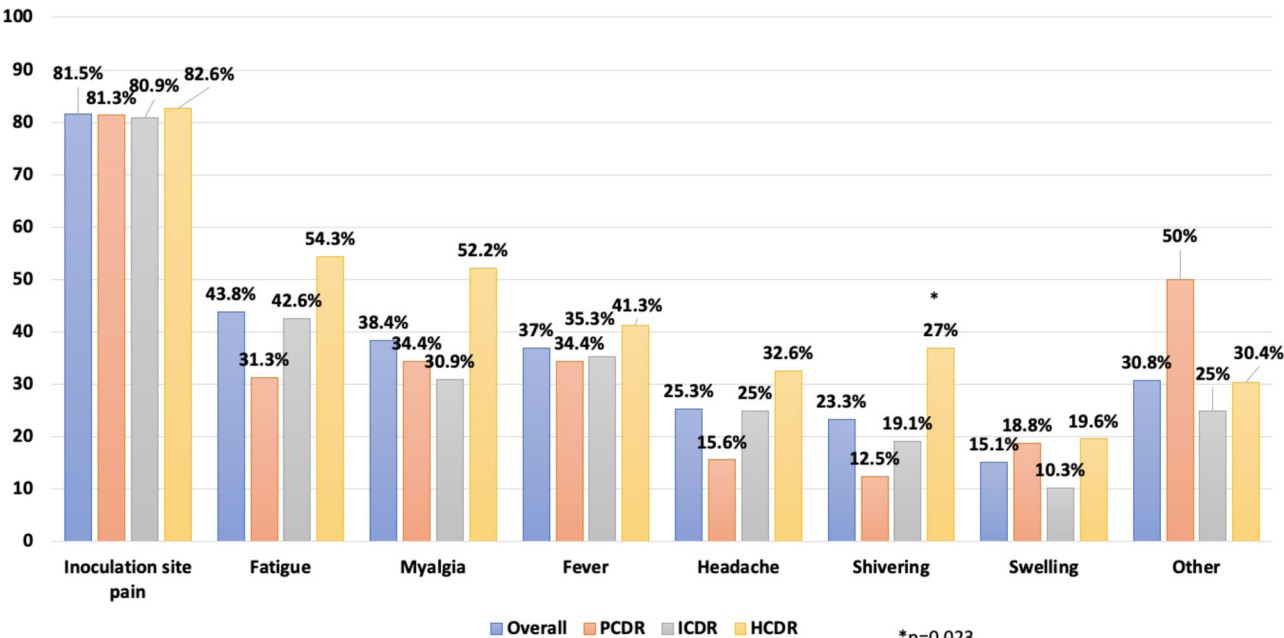

**Fig. 5 | Overall self-reported solicited local and systemic reactions in days 0–7 after vaccination with third dose by CD4 count.** PCDR Low CD4 count recovery, ICDR Intermediate CD4 count recovery, HCDR High CD4 count recovery. Source data are provided as a Source Data file.

and immune-senescence processes[35]. The achieved responses are particularly impressive, considering the participants' status of chronic immune dysregulation, as well as the fact that depletion of viral-specific T and B cell clones is observed even in PLWH responder to antiretroviral therapy (cART)[36,37]. Despite the fact that cART durably leads to HIV-RNA suppression and CD4 + T-cell restoration, and reverses HIV related immune dysfunction[38,39] a persistent immuno-pathology, as that observed in HIV chronic disease, can weaken the general immune responses to vaccination[40,41]. Our cohort of advanced PLWH previously stimulated by a full cycle vaccination showed an optimal response to a third dose. This finding seems to imply that effective cART is able to both inhibit HIV replication and to restore a good immune response, also in patients with low CD4 count or who experienced AIDS events. However, the impact of HIV-related immu-nosuppression on the duration of the vaccine-induced immunity still remains to be fully elucidated. Indeed, it has been reported that the CD4 count is the earliest and critical driver in orchestrating an optimal

vaccine induced immune response[28,30,41]. In chronic PLWH, the CD4 T cells as well as the persistent low inflammatory environment could dampen the differentiation of effective and long-lived memory B and T cell immunity[42,43]. Although overall the data carried little evidence for an association between current level of CD4 count and response to vaccine, we found a significant difference in humoral response when comparing participants with poor CD4 count recovery on ART (PCDR) with those having a high CD4 count recovery (HCDR). Also for the other responses there was a tendency for those with PCDR to show worse outcomes then those having HCDR, although results were not significant likely because of low statistical power. Further studies are mandatory to address this point, which is key to better design the future vaccination schedules in PLWH.

Our study also strongly supports a difference in terms of humoral response to the third dose depending of the type of vaccine sequence received: participants who received as primary series BNT162b2 fol-lowed by a third dose with mRNA-1273 showed a higher mean increase

of humoral response than those received other sequences. Although formal conclusions regarding the clinical significance of this observed difference cannot be made, our data are consistent with the results of another recent study showing higher immunogenicity and significantly enhanced effectiveness of two doses of mRNA1273 compared with BNT162b2[44,45] and with the hypothesis that the observed higher vaccine response with the mRNA-1273 could be related to the additional week between administrations, higher vaccine dose, differences in inducing T cell subsets, or other factors[30].

Further, a blinded, multicenter, randomized, controlled, phase 2 trial performed in the UK (the COV-BOOST study), have shown that among the type of vaccine typically used as booster dose after a primary cycle with BNT162b2, mRNA127 was the type of vaccine able to elicit the highest titres of anti-Spike IgG, neutralizing wild type virus antibodies and cellular immunity, with comparable results regardless of age[46].

Our analysis also provides indirect evidence that the strong neutralizing activity achieved with a third dose may reduce the risk of infection with emerging VoC, such as the Omicron variant. Indeed, recent in vitro studies have demonstrated that higher level of nAbs titres, enhancing a cross reactivity of antibody response, are needed to neutralize Omicron, which could be obtained by deploying third booster doses of vaccine[8–11,47]. A mRNA third dose seems to enhance overall levels of anti-SARS-CoV-2 neutralizing antibodies, above a threshold allowing inhibition of Omicron[13], even though the waning of this immune response has not been yet fully recognized[10].

Moreover, the T cell response induced by both vaccination and natural infection has been shown able to efficiently recognize the Omicron variant (70−80% of CD4 and CD8 T cells cross-recognize Omicron), thus contributing to the protection from severe COVID-19. Therefore, the vaccination in PLWH could significantly reduce the risk of severe Omicron disease by increasing the titer of neutralizing antibody and by inducing an efficient and cross-reactive T cell response[48,49].

Finally, the safety profiles after the additional injection were generally similar to those observed after the primary series and reported in the previously reported phase 2 and 3 clinical trials[50–52]. The most common local side effect was pain in the inoculation site while, among systemic events after the third dose, were fatigue, headache, myalgia and fever which occurred at similar frequencies. These frequencies are similar to those reported in randomized trials although the COV-BOOST trial estimates are significantly higher for those who received as primary series BNT162b2 and for booster dose mRNA-1273[46].

To our knowledge, ours is the first analysis to characterize the immune response and reactogenicity to a third dose of SARS-CoV-2 vaccination in PLWH, and to provide quantification estimates of the level of immunogenicity elicited by an additional boosting dose in PLWH, particularly in those with advanced immune dysregulation. Our data also documents how the level of response may differ by mRNA COVID-19 vaccines administered as third dose according to the primary cycle used and which sequence has the best chance to boost serologic, neutralizing and cell-mediated response in this vulnerable population.

The main limitation of our analysis is the observational, not randomized nature of the study design, so that uncontrolled sources of unmeasured confounding bias (e.g. socio-economic factors) may exist[16]. Despite the prospective design and the careful methodological evaluation, causal links are difficult to establish in this setting. This is particularly true for the association between CD4 count at $T_0$ and the variation in response between $T_{-1}$ and $T_1$ because the evaluation of the exposure does not strictly precede the outcome. The high proportion of participants with a truncated value for anti-RBD and nAbs responses at $T_1$ also further complicated the interpretation of the results for these outcomes.

Also, the definition of the binary outcome for the T-cell response was based on a non-standardized threshold and because of the small number of events observed especially in the ICDR and HCDR groups, the evaluation of the association with current CD4 count with all binary endpoints was likely to be underpowered.

Moreover, the present analysis concerns only the short term response (up to 15 days after the 3D) and therefore we were unable to provide estimates of the durability of immune response and waning of immunogenicity after a third dose in PLWH.

A further limitation is the fact that clinical outcome such as the rate of infection and of disease severity was not evaluated so we were unable to establish whether the increased level of immunogenicity was associated also with a reduced incidence of these clinical events. Although the concept is supported by in vitro data and by studies conducted in the general population[3–11], whether the high level of nAbs elicited by the third dose in our population is able to reduce the risk of infection with newly circulating VoC such as Omicron remains to be seen.

Despite these limitations, the findings presented here indicate that in PLWH with advanced disease at the time of their first vaccination dose, providing an additional dose of SARS-CoV-2 mRNA vaccine at least 4 months after the initial two-dose vaccination, resulted in markedly higher levels of boosted immunity after initial course. Although the clinical effectiveness of the third dose has to be definitively proven by further and larger studies, these early data appears to support the decision to provide a short term third dose to this subset of PLWH.

## Methods
### Study design and population
On September 10th, 2021, as part of the Nationwide Booster Vaccination Program in Italy, the National Institute for Infectious Diseases Lazzaro Spallanzani in Rome started the boosting vaccination against SARS-CoV-2 in PLWH, as a third dose of a mRNA vaccine (a full additional dose of BNT162b2 or mRNA-1273), according to the Italian Ministry of Health recommendations, for those who, at the time of their first vaccine dose showed a CD4 < 200/mm³ or were previously diagnosed with AIDS.

Participants in this analysis are a subset of those who, following written informed consent, had been enrolled in an observational cohort study to evaluate the outcomes of SARS-CoV-2 vaccination (the HIV-VAC study). HIV-VAC was approved by the Scientific Committee of the Italian Drug Agency (AIFA) and by the Ethical Committee of the Lazzaro Spallanzani Institute, as National Review Board for COVID-19 pandemic in Italy (approval number 423/2021). Details of this study have been described elsewhere[27].

Specifically, the present analysis includes PLWH consecutively enrolled in HIV-VAC for whom in October 20th, 2021 there was at least a 28 days gap after having completed the 2-dose schedule with BNT162b2 or mRNA-1273 vaccines, and receiving a third dose. Individuals with a SARS-CoV-2 infection diagnosis, defined by a RT-PCR positive to the molecular test on the nasopharyngeal swab, or positivity to anti-N IgG at $T_0$, were excluded for the present analysis.

Participants' demographic, epidemiologic, clinical and laboratory characteristics at time of the third dose were collected. Exact vaccine sequence was recorded. Also, humoral and neutralizing antibodies responses were retrospectively measured in blood samples which were stored at time of third dose ($T_0$) and approximately 15 days after the third dose ($T_1$). T-cell response was measured on fresh blood collected at the same times. In addition, both humoral neutralizing antibodies and T-cell responses measured approximately 30 days after participants' second vaccine dose were also available for comparative analysis (time $T_{-1}$). In addition, at day 7 after the third dose, participants were asked via a telephone interview about solicited adverse events which might have occurred over the period 0−7 days following the

third dose. Finally, ninety-eight HIV seronegative health care workers (HCWs) were included for comparison, from whom samples were collected before and after the booster dose (to be intended as a booster dose, a full dose of BNT162b2 or half a dose of mRNA-1273).

## Laboratory procedures

Two commercial chemiluminescence microparticle antibody assays (CMIA), the SARS-CoV-2 specific anti-N, and the anti-S/RBD tests (ARCHITECT SARS-CoV-2 IgG, and ARCHITECT SARS-CoV-2 IgG II Quantitative, Abbott Laboratories, Wiesbaden, Germany respectively,) were performed on ARCHITECT® i2000sr (Abbott Diagnostics, Chicago, IL, USA) and used according to manufacturer's instruction; Index >1.4 and Binding Antibody Units (BAU)/mL ≥ 7.1 are considered positive, respectively.

Micro-neutralization assay (MNA) was performed using SARS-CoV-2/Human/ITA/PAVIA10734/2020, as challenging virus[16]. Briefly, serum samples were heat-inactivated at 56 °C for 30 min, and titrated in duplicate in 7 two-fold serial dilutions (ranging from 1:10 to 1:640). Equal volumes (50 µL) of serum and medium containing 100 tissue culture infectious doses 50% ($TCID_{50}$) SARS-CoV-2 were mixed and incubated at 37 °C for 30 min. Serum-Virus mixtures were then added to sub-confluent Vero E6 cell (ATCC, Manassas, Virginia, United States, CRL-1586™) monolayers and incubated at 37 °C and 5% $CO_2$. After 48 h, microplates were observed by light microscope for the presence of cytopathic effect (CPE). To standardize inter-assay procedures, positive control samples showing high (1:160) and low (1:40) neutralizing activity were included in each assay session. Serum from the National Institute for Biological Standards and Control, UK (NIBSC) with known neutralization titer (Research reagent for anti-SARS-CoV-2 Ab NIBSC code 20/136) was used as reference in MNA. The standardized cut off of $MNA_{90} \geq 1:10$ was used to define neutralization activity; only for computational and statistical purposes, samples resulted > = 1:640 were arbitrarily considered =1:1280.

We studied IFN-γ production in response to Spike stimulation as a surrogate of specific T-cell function. Peripheral blood was collected in heparin tubes and stimulated or not with a pool of peptides spanning the Spike protein (Miltenyi Biotech, Germany) at 37 °C (5% $CO_2$). A superantigen (SEB) was used as positive control. Plasma were harvested after 16-20 hours of stimulation and stored at −80 °C. IFN-γ released in plasma after stimulation was quantified using an automated ELISA (ELLA, Protein Simple). The detection limit of these assays was 0.17 pg/mL for IFN-γ and the cut off used in this analysis to define the T-specific cells response was 12 pg/mL, calculated as the mean + 2 SD of the response to spike peptides of unvaccinated uninfected heathy donors[53].

## Statistical analysis

The primary outcome was immunogenicity (humoral, neutralizing and cell-mediated responses) measured 15 days after receiving the third dose ($T_1$). This was defined as both quantitative/continuous (average difference between $T_0$ and $T_1$) or qualitative/binary (lack of response). The latter outcomes were immunogenic parameter specific which were defined as follows: anti-RBD IgG Binding Antibody Units <7.1 (BAU)/mL, $MNA_{90}$ < 1:10 and for IFN-γ < 12 pg/mL. Secondary outcomes were i) 0−7 days reactogenicity (self-reported by telephone interview), and ii) the difference in average level of immunogenicity between $T_{-1}$ and $T_1$.

Because the distribution of the immunogenic response parameters was positively skewed, a $log_2$ transformation was used for all measures (RBD-binding IgG, nAb titres, IFN-γ), to make the data conform more closely to the normal distribution and to improve the model fit.

Mean and standard deviation in the $log_2$ scale are presented and a paired t-test was used to test a difference from zero in the overall changes over $T_0$−$T_1$.

Participants were then stratified by CD4 count at $T_0$ in three groups according to the size of CD4 count recovery: CD4 count <200 cell/$mm^3$: poor CD4 recovery (PCDR); CD4 count between 200 and 500 cell/$mm^3$: intermediate CD4 recovery (ICDR); CD4 count >500 cell/$mm^3$: high CD4 recovery (HCDR) and according to vaccine sequence: i) primary series with BNT162b2, third dose with BNT162b2 (BNT162b2 group); ii) primary series with mRNA-1273, third dose with mRNA-1273 (mRNA-1273 group); iii) primary series with BNT162b2, third dose with mRNA-1273 (2BNT162b2 + mRNA-1273 group); iv) primary series with mRNA-1273, third dose with BNT162b2 (2mRNA-1273 + BNT162b2). Strata specific mean values of RBD-binding IgG, nAb titres, IFN-γ with standard deviations were also shown.

Proportions of participants who failed to achieve a level of immunogenicity above the thresholds described above were calculated by CD4 count and vaccine sequence strata and compared using a chi-square or Fisher exact test as appropriate. ANOVA, logistic regression models and truncated regression models were used to evaluate the association between these two exposure factors (current CD4 count and vaccine sequence) and the level of immunogenic response.

In the ANOVA analysis we used the naïve approach of replacing the truncated values with the upper limit of the assay for anti-RBD > 7.1 BAU/mL and nAbs > 1:10). When comparing mean responses by ANOVA, after checking that an overall difference between the groups existed, specific pairwise contrasts hypothesis testing was performed, after controlling for the inflation of type I error due to multiple testing. Specifically, Dunn's test with Bonferroni correction was used to adjust the p-values of these pairwise contrasts.

Univariable and multivariable regression models were fitted (logistic and truncated regression). For the logistic regression the binary outcome was lack of immunogenic response as defined above. Because of the large number of participants with a response value above the upper limit cut-off of the assay, ANOVA results could be biased. Truncated linear regression adequately controlled for censored data for the outcome variable, it was similar to ANOVA but was fitted on the natural scale of the responses and correctly accounted for truncated values (participants who reached the upper limit of the assay for specific responses).

When CD4 count at $T_0$ was the exposure of interest, the following variables have been identified as potential confounders for the association between CD4 count and immunogenic responses: age, time from HIV diagnosis, CD4 nadir, HIV-RNA at the time of third dose, days from the date of $2^{nd}$ dose, vaccine sequence and concomitant cancer. These model assumptions are described by means of a direct acyclic graph (DAG), built using DAGitty vers. 2.3 released 2015-08-19, available at http://www.daggity.net/ (Supplementary Fig. 1). Because vaccine sequence allocation was pseudo-random, only age-adjusted models have been used to evaluate the association between this factor and immunogenic response.

Furthermore, we performed an additional analysis of covariance analysis with the aim to compare humoral and SARS-CoV-2 T-cell specific between our cohort of PLWH and health care workers (HCWs) after controlling for gender, age and time difference in the exact day intervals between the booster dose and the measured response (between $T_0$ and $T_1$ and between $T_{-1}$ and $T_0$).

A descriptive analysis of the secondary outcome measuring self-reported side effects was also performed showing the proportion of participants reporting specific side effects by CD4 count and vaccine sequence strata. Chi-square test was used to compare these proportions.

All statistical analyses were performed using SAS Statistical Software v.9.4 (SAS Institute Inc., Carey, NC, USA). All figures were generated using GraphPad Prism 9.0 (GraphPad Software, Inc., San Diego, CA).

## Reporting summary

Further information on research design is available in the Nature Research Reporting Summary linked to this article.

## Data availability

The HIV-VAC data are available under restricted access for confidentiality reasons, since these patients may be identified by combinations of person-specific characteristics within the database; access can be obtained by specific request to the corresponding author. The raw data on demographics and clinical status of participants, are protected and not available due to data privacy laws. The processed data are available by specific request to alessandra.vergori@inmi.it. Source data are provided with this paper.

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

## Acknowledgements

The authors gratefully acknowledge nurse staff, all the patients and all members of the HIV-VAC Study Group.

## Author contributions

A.A., A.V., and S.L. conceptualized, designed the study, and wrote the protocol. A.V. and A.C.L. wrote the first draft of the manuscript and referred to appropriate literature. A.C.L. was also the main responsible person for formal data analysis. A.A., A.C.L., A.V., C.A., C.C., F.V., and E.G. conceived, supervised the study, and contributed to data interpretation. S.C. and L.D.P. were responsible for data curation; V.P. is the main responsible of data on immunogenicity on HCWs and revised the manuscript; V.M., I.M., A.R.G., S.C. revised the manuscript content, reviewed and edited the manuscript. G.M., S.M., and F.C. performed all the serology tests and neutralization assays; V.B., E.C., and D.M. performed all the T-cell function tests; P.G., and A.M. performed and supervised the anti-SARS-CoV-2 vaccination campaign at INMI for HIV-positive individuals; R.I. and LDP enrolled participants. All authors agreed with and approved the final version of the manuscript.

## Funding

The study was performed in the framework of the SARS-CoV-2 surveillance and response program implemented by the Lazio Region Health Authority. This study was supported by funds to the Istituto Nazionale per le Malattie Infettive Lazzaro Spallanzani IRCCS, Rome (Italy), from

Ministero della Salute (Programma CCM 2020; Ricerca Corrente – Linea 1 e Linea 2; COVID-2020-12371675).

## Competing interests
The authors declare no competing interests.

## Additional information

## HIV-VAC study group

Chiara Agrati[4], Alessandra Amendola[3], Andrea Antinori[1], Francesco Baldini[1], Rita Bellagamba[1], Aurora Bettini[1], Licia Bordi[3], Veronica Bordoni[4], Marta Camici[1], Rita Casetti[4], Concetta Castilletti[3], Stefania Cicalini[1], Francesca Colavita[3], Sarah Costantini[1], Flavia Cristofanelli[4], Alessandro Cozzi Lepri[2], Claudia D'Alessio[1], Veronica D'Aquila[1], Alessia De Angelis[8], Federico De Zottis[1], Lydia de Pascale[1], Massimo Francalancia[3], Marisa Fusto[1], Roberta Gagliardini[1], Paola Galli[5], Enrico Girardi[7], Giulia Gramigna[3], Germana Grassi[4], Elisabetta Grilli[1], Susanna Grisetti[1], Denise Iafrate[1], Roberta Iannazzo[1], Simone Lanini[1], Daniele Lapa[3], Patrizia Lorenzini[1,10], Alessandra Marani[4], Erminia Masone[1], Ilaria Mastrorosa[1], Davide Mariotti[4], Stefano Marongiu[8], Giulia Matusali[3], Valentina Mazzotta[1], Silvia Meschi[3], Annalisa Mondi[1], Stefania Notari[4], Sandrine Ottou[1], Jessica Paulicelli[1], Luca Pellegrino[8], Carmela Pinnetti[1], Maria Maddalena Plazzi[1], Adriano Possi[9], Vincenzo Puro[6], Alessandra Sacchi[4], Eleonora Tartaglia[4], Francesco Vaia[5] & Alessandra Vergori[1,11]

[8]Department of Health Professions Nursing, National Institute for Infectious Diseases Lazzaro Spallanzani IRCCS, Roma, Italy. [9]Vaccine Center Administrative Coordination, National Institute for Infectious Diseases Lazzaro Spallanzani IRCCS, Roma, Italy. [10]Present address: National Center for Disease Prevention and Health promotion, National Institute of Health, Rome, Italy.

