## [Peer Review File · Nature Communications]

Immunogenicity to COVID-19 mRNA vaccine third dose in people living with HIVReviewers' Comments:

Reviewer #1:

Remarks to the Author:

Immunogenicity and reactogenicity to COVID-19 mRNA vaccine additional dose in people living with HIV

Vergori A

Nat Com

Host factors associated with reduced vaccine efficacy and durability are well established and include older age in adults, chronic inflammation, CMV latent infection and immunosuppression. These factors hamper vaccine response with limited elicitation of antibody titers and reduced T-cell response, in relation with low thymic output and a restricted naïve T-cell repertoire.

Information on COVID-19 vaccine immunogenicity and immune response durability in people living with HIV (PLWH) is not yet established, as these persons were mostly excluded from large perspective vaccine studies.

PLWH represent a distinctive group among immunosuppressed patients. HIV infection is characterized by a reduction of the number of CD4 T-cells, predominantly the mucosal protective CD4 T-helper 17 (Th17) cells that are rapidly infected and depleted from the digestive track and to a lesser extent from the respiratory track. As cell-mediated immunity, including CD4 helper T-cells, has emerged as a critical contributor to the COVID-19 vaccine response, the influence of combined immunosuppression and immune activation in PLWH remains a poorly addressed question.

Some reports showed similar immunogenicity in ART treated with good CD4 PLWH and health care controls.

Lombardi A. Anti-spike antibodies and neutralising antibody activity in people living with HIV vaccinated with COVID-19 mRNA-1273 vaccine: a prospective single-centre cohort study. *Lancet Reg Health Eur.* 2021 Dec 23:100287.

However, we must address immunogenicity in poorly control PLWH. Vergori et al. assessed immunogenicity after third dose in 216 advanced patients with CD4 count <200 cell/mm³ and/or previous AIDS at the time of their first COVID vaccine dose. The study started on September 10, 2021, the administration of an additional (third) dose (AD) of anti-SARS-CoV-mRNA vaccine was approved in Italy to be given after >28 days after completion of the primary vaccination cycle in PLWH who presented with a CD4 T cell count <200/mm³ and/or previous AIDS at the time of their first dose.

Results are very timely and showed absence of boosting effect on cell-mediated immunity and was encouraging as humoral response was higher than the one of the second dose and even better if a mRNA heterologous combination was used as mRNA-1273 as third dose.

Study findings are timely and novel for advanced PLWH and are encouraging for the humoral response after 3 doses.

Major comments:

Lack of control group of age match health care workers limiting the validation of immune markers used in this work, and for potential inference for vaccine effectiveness.

The study did not address vaccine effectiveness and durability of humoral, nor cellular immune response, key factors for real life vaccine success. Alrubayyi et al. showed in London, UK, that antibody and specific T-cell responses to SARS-CoV-2 infection were comparable between PLWH and negative subjects and persist 5-7 months following mild COVID-19 disease. Importantly, the magnitude of specific T-cell responses was associated with the naïve CD4 T-cell pool size and the CD4/CD8 ratio in PLWH. Moreover, PLWH with CD4 T-cell below 200 cells/ μ L, a threshold for increased risk of AIDS, may not respond well and durably to vaccine, as previously reported for other vaccines.

Vaccine response might, in any population, wanes due to developments of variants escaping T-cells, Spillover and spillback of SARS-CoV-2 variants from human to animal then back and antibody selective immune pressure (i.e.: antigenic drift and to short-lived immune memory for SARS-CoV-2, which is not a virus with obligate viremic spread. Therefore, the today relevance for delta and omicron variants remains uncertain, limiting the clinical and public health relevance of the study findings.

Alrubayyi A, Gea-Mallorqui E, Touizer E, Hameiri-Bowen D, Kopycinski J, Charlton B, et al. Characterization of humoral and SARS-CoV-2 specific T cell responses in people living with HIV. *Nat Commun.* 2021;12(1):5839.

Yewdell JW. Antigenic drift: Understanding COVID-19. *Immunity.* 2021;54(12):2681-7.

Yewdell JW. Individuals cannot rely on COVID-19 herd immunity: Durable immunity to viral disease is limited to viruses with obligate viremic spread. *PLoS Pathog.* 2021;17(4):e1009509.

Costiniuk CT, Singer J, Langlois MA, Kulic I, Needham J, Burchell A, et al. CTN 328: immunogenicity outcomes in people living with HIV in Canada following vaccination for COVID-19 (HIV-COV): protocol for an observational cohort study. *BMJ Open.* 2021;11(12):e054208.

Abstract and introduction:

AD additional dose is not clear, please replace by third dose (3D) for vaccine throughout the text and abstract

Introduction: Introduction is too long to general as now several paper have been reported should immediately start with vaccine response in PLWH.

These data are consistent with the observation that HIV infection may favor a poor serological response also to vaccines for viral agents, such as influenza. You need to add that low socioeconomic factors, professional exposition to COVID are also associated with PLWH, including co-morbidity that may be associated of more severe outcome, independently of immune status.

Fehr D. Characterization of people living with HIV in a Montreal-based tertiary care center with COVID-19 during the first wave of the pandemic. *AIDS Care.* 2021 Mar 28:1-7

SID: <200/mm³; minor immunodeficiency, MID: 200-500/mm³; no immunodeficiency, NID: 171 >500/mm³. These abbreviations are confusing and it will be easier for readers to directly call groups less than 200, 200-500 and more than 500. It is incorrect to speculate that more than 500 equal no immunodeficiency in HIV. Change in the text and in table 1 and should be consistent with Cd4 groups in table 2.

Contribution of CD4 CD8 ratio, should be presented as for some antibody vaccine response a better predictor than absolute CD4 counts.

BNT162b2 (homologous P, HP); ii) primary series with mRNA-1273, third additional dose with mRNA-1273 (homologous M, HM); iii) primary series with BNT162b2, additional dose with mRNA-1273 (heterologous M, HtM); iv) primary series with mRNA-1273, additional dose with BNT162b2 (heterologous P, HtP). Strata specific mean values of RBD-binding IgG, nAb titres, IFN- γ with standard deviations were also shown. This mixture of vaccine sequences is hard to read and to keep in mind the numerous abbreviations proposed and then lower in the text. This part should be limited and presented as supplemental data, too difficult to follow in the current presentation.

Minor comments:

Reactogenicity should be reduced to a global commentary as the text is too long, no this chapter does not bring any new data. As reactogenicity is not generating new data, the term should be removed from the title and shortly discussed and presented data in a supplemental figure or table only.

Discussion is much too long and redundant with previous reports referred by investigators. Currently, several reports already focused on PLWH vaccine response. The discussion should mainly focus on PLWH with more than 350 and less than 200, which is the novelty of this paper.

Throughout the text use the same term immunogenicity, humoral response, boosted immunity after initial course.

Globally the text is not very clear and should be shortened with the focus on below CD4 T cell 200, which is the novelty here.

Reviewer #2:

Remarks to the Author:

Vergori and colleagues investigated the safety and immunogenicity of an additional (third) vaccine dose for PLWH. The PLWH are further stratified based on CD4 count, which is used as a proxy for disease severity and efforts were undertaken to evaluate whether disease severity impacts on the response to the booster vaccine.

An important motivation for this study was the question whether PLWH, considered an immune-suppressed population, would benefit from a booster vaccine. This is definitely a justified question, but has lost some relevance as many governments by now have decided on providing booster vaccines to the entire adult population. Nonetheless, investigating whether whether the standard mRNA vaccination regimens are also effective for the population of PLWH and whether subsets of PLWH in terms of vaccine response exist remains a valid question.

Although the experiments reported in this study are overall well-performed, the study has several important limitations most of which are in fact identified by the authors in the discussion. Overall, this paper mainly confirms prior reports of good immune responses to SARS-CoV-2 vaccines in the global population of PLWH and shows that the 3rd dose, as in the general population, further enhances immunity in the majority of PLWH. A major limitation is the fact that the study was underpowered for the ambitious aims of 1) identifying subpopulations of PLWH that would have suboptimal vaccine responses and 2) identifying optimal vaccination regimens for this subset. Also, the fact that the study did not contain a healthy control group makes it difficult to assess the extent to which the response of the PLWH cohort to the AD differs from that of the general population. Taken together, although several interesting trends are reported, drawing firm conclusions from this study is difficult.

Specific comments

- Abstract line 36: "we analyzed anti-RBD, microneutralization assay and IFN γ production in 216 PLWH on ART with advanced disease (CD4 count <200 cell/mm³ and/or previous AIDS) at the time of their first dose": it is not clear to me how this population is then stratified at T0 in the severe immunodeficiency (SID: <200/mm³); minor immunodeficiency (MID: 200-500/mm³) and no immunodeficiency, (NID: >500/mm³) groups that are used for statistical analyses. What are the PLWH with advanced disease when they received their first vaccine dose that end up in the MID and NID groups at T0? Are the CD4 counts that variable between the first vaccine dose and the AD? In that case this does not seem like a very good parameter to use for stratification. From Table 2 it does seem that the MID and NID groups are enriched for PLWH with prior AIDS. Is it correct to assume then that these have higher CD4 counts? That seems a bit counterintuitive... Are the CD4 counts at time of first dose known for the patients with prior AIDS? Perhaps a figure with the evolution of CD4 counts from first vaccine dose to T0 can be included for all 216 participants?

- Line 153: "The detection limit of these assays was 0.17 pg/mL for IFN- γ and the cut off used in this analysis to define the T specific cells response was 12 pg/mL." What was the rationale for selecting this cut-off? Likewise: how were the cut-offs for other assays determined? For the Nab dilution it is stated that the cut-off is 1:128 (line 253). Is this correct? Some points in the graphs have a log₂ dilution over 10 (hence dilution of over 1000?). In the mat & meth section (line 138) it is stated that "7 two-fold serial dilutions (starting dilution 1:10)" was used which would indicate that the highest dilution tested would be 1:640 if I am correct? Can this be clarified?

- It would be helpful to show the cut-offs used for qualitative statistical analyses on the graphs.

- Fig. 1-3: the use of "increase" on the title is confusing as this suggest a delta value is represented which is not the case and moreover some individuals show a decrease. Also, for Fig. 1 the use of "peak" in the title is inappropriate as there are no longitudinal data showing that the sampling time point indeed represents the peak response.
- Line 181: "In the ANOVA analysis we used the naïve approach of replacing the truncated values with the upper limit of the assay for anti-RBD >7.1 BAU/mL and nAbs >1:10". Can you clarify why this approach was chosen.
- Line 189: "truncated regression analysis was similar to ANOVA but was fitted on the natural scale of the responses and correctly accounted for truncated values (participants who reached the upper limit of the assay for specific responses)." Do you mean that no log transformation was done here?
- Line 237: There is an association between IgG titers and CD4 counts but not neutralization capacity and CD4 counts. Do IgG and neutralization titers correlate?
- The discussion focuses on comparison to other immune deficient populations such as individuals taking rituximab or chemotherapy. This might not be the most relevant comparison. It would be more interested to discuss what is known on the immune dysfunction that is characteristic for PLWH an how this impacts vaccination.
- The authors state in the discussion that "The achieved responses are particularly impressive, considering the participants' status of chronic immune activation, as well as the fact that depletion of viral-specific T and B cell clones is observed even in PLWH responder to antiretroviral therapy (cART)". Is there information on the inflammatory status of the study participants?

Minor

- The manuscript still contains quite a bit of grammatical errors: use of articles, singular-plural mistakes. Also, some sentences are formulated in a way that makes them difficult to grasp the message. Perhaps an additional round of language editing would be adviseable.
- Although this is clearly described in the Methods, from the abstract it is difficult to understand what is meant by the "additional dose" (AD). Perhaps it can already be clearly stated in the abstract that the additional dose is the third vaccination given to individuals that have completed the standard two-dose mRNA vaccination regimen (either Pfizer or Moderna). And that the investigated population includes both recipients of a homologous and heterologous AD.
- Although the authors are right that no reports on response to third doses in PLWH are available, some additional studies on responses of PLWH to the standard regimen, could be referenced (eg. Jedicke et al. Humoral immune response following prime and boost BNT162b2 vaccination in people living with HIV on antiretroviral therapy. *HIV Med.* 2021 Nov 2;10.1111/hiv.13202. doi: 10.1111/hiv.13202. Online ahead of print. PMID: 34725907; Spinelli et al. Differences in Post-mRNA Vaccination SARS-CoV-2 IgG Concentrations and Surrogate Virus Neutralization Test Response by HIV Status and Type of Vaccine: a Matched Case-Control Observational Study. *Clin Infect Dis.* 2021 Dec 5:ciab1009. doi: 10.1093/cid/ciab1009. Online ahead of print. PMID: 3486496; Aledo et al. Safety and immunogenicity of SARS-CoV-2 mRNA-1273 and BNT162b2 vaccines in people living with HIV. *AIDS.* 2022 Jan 6. doi: 10.1097/QAD.0000000000003161. Online ahead of print. PMID: 34999608)
- Line 53: nevertheless? Seems like "in addition" or something alike would be better suited here?
- Line 59: "the wild-type D614G virus"; perhaps rename as G614 SARS-CoV-2 variant
- Line 67: "Several studies recently supported a significant increase of neutralization against Omicron after a booster dose, even lower than that observed with ancestral type or Delta suggesting a cross-reactivity of neutralizing antibody responses": do you mean that the increase in neutralization is lower?
- Line 114: "consecutive PLWH": what do you mean by this?
- Line 116: "Individuals with a SARS-CoV-2 infection diagnosis, defined by a RT-PCR positive to the molecular test on the nasopharyngeal swab, or positivity to anti-N IgG, were excluded for the present analysis." Were these analyses conducted on samples from T0 and T1?
- Line 123: "In addition, both humoral neutralizing antibodies and T-cell responses measured approximately 30 days after participants' second vaccine dose were also available for comparative analysis (time T-1)." Is the reason that these were not available for all participants that a major subset had a smaller gap between dose 2 and dose 3 (AD)?

- Line 226: "The humoral response elicited by AD was on average stronger than the peak titres elicited by the primary vaccination 1 month after the completion of 2 doses vaccination cycle" : is it appropriate to say that the response to the AD is stronger as different timepoints post-vaccination are compared?
- Line 336: Are nasal congestion and heavy breathing not unusual vaccine side effects? Where these reported in the recent study by Munro et al. that is also cited in the discussion?
- Line 393: "The T cell contraction observed overtime can be effectively reverted by the additional dose, with some efficacy as reported by the primary vaccination schedule": What do you mean by this?
- Table 1: the title (current CD4 count) is confusing and not necessary. The readability of the table could be improved by using a format where table cells have borders.
- Table 2: why not replace CD4 T cell counts by SID, MID, NID?
- Line 452: "These prevalences...": which prevalences

Reviewer #3:

Remarks to the Author:

Vegori et al characterize humoral and cellular responses to a third dose of COVID-19 mRNA vaccine in individuals living with HIV who had met the definition of advanced disease (CD4 count <200 cells/mm³ or prior AIDS diagnosis) at the time of their first COVID-19 vaccine dose. The team also characterize reactogenicity. The results are timely, as there is presently little information on immune responses to 3rd COVID-19 vaccine doses in people with HIV. The cohort size (N=216), the fact that there are two historic longitudinal datapoints following the 2nd dose for most participants, and the fact that the researchers measured both humoral and cellular responses, are additional strengths of the study.

However, I have a number of major concerns, many that relate to unclear descriptions/justifications of the statistical analyses performed, that substantially dampen my enthusiasm for the manuscript.

1. Data are presented in a redundant manner. For example, the immune response data from the T1 timepoint are presented in Figure 1 (where these values are compared to T-1), Figure 2 (where these values are compared to T0), Figure 3 (same data as Figure 2 but stratified by CD4 count) and Figure 4 (essentially same data as Figure 2 but stratified by vaccine regimen), Supplemental Figures 2-4 (same data as Figure 1 but stratified by CD4 count), with separate multivariable analyses adjusting for CD4 count and vaccine regimen, respectively, presented in Tables 2 and 3. The authors should instead clearly delineate the paper's main goal(s)- e.g. to characterize the relationship between CD4 count and post-3rd dose responses in PLWH - and present these data in a succinct set of figures, with a single multivariable model that adjusts for other possible confounders including vaccine regimen.

2. Statistical analyses are incompletely described and justified. Some examples are:

a) Use of a binary cutoff set at each humoral assay's lower limit of quantification (LLOQ) is not appropriate for a post-3rd dose vaccine immunogenicity study as the number of nonresponders is essentially zero (e.g. there are only 3 nonresponders out of all samples shown in Figure S2). If a binary cutoff remains appropriate for the cellular immune analyses, the value (12 pg/mL) should be justified.

b) lines 193-197. How did the authors select the variables to include in the DAG analysis, and why was this approach chosen? Table 1 for example suggests additional variables that could serve as confounders (e.g. liver disease, dose timing), why were these not included? Related to this, what factors were included in the multivariable models presented? Lines 198-199 indicate that only age adjustment was performed, but the text (e.g. lines 292-296) lists additional variables for certain analyses, though not others.

c) The comparisons performed in Figures S2-S4 are not clear. How were these data reduced to frequency summaries needed to apply Fisher's exact or chi-squared tests? What data were compared to generate the p-value in the title of each figure, and how does this differ from the comparisons that were performed to generate the p-values above the figure panels? Related to this, in Figure S3, how can the p-value in the title be highly significant when none of the other comparisons are significant? The same issue affects Figure 3. As mentioned above, binary cutoffs are not appropriate for the humoral data anyway. But, part of the confusion here may be that the figure highlights a change in response between two timepoints, presented on a continuous scale, when the authors may in fact be converting results from only one of these two timepoints to a binary outcome, and comparing those frequencies in 2x2 tables (?)

d) lines 275, 282, 296: on both these lines, the authors state that a multivariable ANOVA was performed, and then report a p-value for a Fisher's exact test. Are the authors instead referring to the model F-statistic? Moreover, when describing multivariable analyses, it is essential to clearly state what outcome is being evaluated, as well as what variables were included in each model. For example, in lines 279-285, the text suggests that the outcome variable is the change between T0-T1 (though this should read T1-T0), but the figure suggests that the outcome variable was the magnitude of the response at T1. Furthermore, the text indicates that the model adjusted for age, but what about other potential confounders?

e) The same question of what data are being compared affects Figures 4A-C. Here, are authors comparing the mean increase between timepoints (as indicated in the figure title) or the responses at T1 (as seems to be indicated in the figure)

f) The authors should use caution if and when analyzing any outcome variable that represents the magnitude of change between a prior timepoint and T1, as a substantial minority of values at T1 (particularly for anti S-RBD IgG) are above the assay limit of quantification, which means that any magnitudes of change for these participants will be underestimates of the true change. As the latter occurs most often in the NID group, this could lead to bias.

g) lines 189-190 (truncated regression analysis) are unclear: what do the authors mean when they say that "ANOVA was fitted on the natural scale of the responses and correctly accounted for truncated values?"

3. Some comments in the discussion appear to contradict the data presented, though this could be due to a lack of clarity, again on what outcome is being discussed. For example:

a) line 351 states that "no significant association could be found with... CD4 count" but this contradicts lines 237-239 "we found an association between the level of CD4 count at T0 and the observed variations in humoral responses from ...[T-1 to T1]" and lines 414-416 "our data suggest that the CD4 count at time of receiving the AD is a potential determinant of the magnitude of the response...."

b) The authors state on lines 422-423: "participants who received as primary series BNT162b2 followed by an AD with mRNA-1273 showed a more robust humoral response than those received other sequences". But, in Figure 4, the highest T1 responses were observed in people who had received three mRNA-1273 doses. It is possible that the authors are referring to the CHANGE in responses between T0/T1, but this is not clear (and please see caveat in comment 2f).

4. The paper could be additionally strengthened with the inclusion of data from a control group without HIV - it appears from the author's prior work that these participants were recruited

5. Given the rapid spread of Omicron worldwide, the paper could be additionally strengthened with

Omicron-specific assays.

Additional comments:

1. Vaccine regimen type should be added to Table 1
2. On all Figures, consider adding the assay lower and upper limits of quantification. If binary cutoffs are retained, consider adding these as well. For univariable analyses, consider reporting median/IQR rather than mean and applying nonparametric statistics, as some of the data appear non-normal even after transformation. Also, put p-values on each panel, not in the figure title.

Rome, 17th March 2022

We here provide a point-by-point reply to the comments and we have incorporated the related changes in the manuscript.

We thank the reviewers for their thoughtful insights, which helped to significantly improve the manuscript.

Reviewer #1 (Remarks to the Author):

Immunogenicity and reactogenicity to COVID-19 mRNA vaccine additional dose in people living with HIV

Vergori A Nat Com

Host factors associated with reduced vaccine efficacy and durability are well established and include older age in adults, chronic inflammation, CMV latent infection and immunosuppression. These factors hamper vaccine response with limited elicitation of antibody titers and reduced T-cell response, in relation with low thymic output and a restricted naïve T-cell repertoire.

Information on COVID-19 vaccine immunogenicity and immune response durability in people living with HIV (PLWH) is not yet established, as these persons were mostly excluded from large perspective vaccine studies.

PLWH represent a distinctive group among immunosuppressed patients. HIV infection is characterized by a reduction of the number of CD4 T-cells, predominantly the mucosal protective CD4 T-helper 17 (Th17) cells that are rapidly infected and depleted from the digestive track and to a lesser extend from the respiratory track. As cell-mediated immunity, including CD4 helper T-cells, has emerged as a critical contributor to the COVID-19 vaccine response, the influence of combined immunosuppression and immune activation in PLWH remains a poorly addressed question.

Some reports showed similar immunogenicity in ART treated with good CD4 PLWH and health care controls.

Lombardi A. Anti-spike antibodies and neutralising antibody activity in people living with HIV vaccinated with COVID-19 mRNA-1273 vaccine: a prospective single-centre cohort study. *Lancet Reg Health Eur.* 2021 Dec 23:100287.

However, we must address immunogenicity in poorly control PLWH. Vergori et al. assessed immunogenicity after third dose in 216 advanced patients with CD4 count <200 cell/mm³ and/or previous AIDS at the time of their first COVID vaccine dose. The study started on September 10, 2021, the administration of an additional (third) dose (AD) of anti-SARS-CoV-mRNA vaccine was approved in Italy to be given after >28 days after completion of the primary vaccination cycle in PLWH who presented with a CD4 T cell count <200/mm³ and/or previous AIDS at the time of their first dose.

Results are very timely and showed absence of boosting effect on cell-mediated immunity and was encouraging as humoral response was higher than the one of the second dose and even better if a mRNA heterologous combination was used as mRNA-1273 as third dose.

Study findings are timely and novel for advanced PLWH and are encouraging for the humoral response after 3 doses.

Major comments:

Lack of control group of age match health care workers limiting the validation of immune markers used in this work, and for potential inference for vaccine effectiveness. The study did not address vaccine effectiveness and durability of humoral, nor cellular immune response, key factors for real life vaccine success. Alrubayyi et al. showed in London, UK, that antibody and specific T-cell responses to SARS-CoV-2 infection were comparable between PLWH and negative subjects and persist 5-7 months following mild COVID-19 disease.

We performed an additional analysis of covariance analysis with the aim to compare humoral and SARS-CoV-2 T-cell specific between our cohort of PLWH and a control group of HIV negative health care workers (HCWs) after controlling for gender, age and difference in the exact days interval between the booster dose and the measured responses (between T0 and T1 and between T-1 and T0). Results are shown in the Table below. PLWH appeared to have not impaired anti-RBD and nAbs responses as compared to HCW after adjustment. In contrast, the data carried a significant evidence for a lower mean difference in IFN-gamma between PLWH and controls. Our data are consistent with those shown by Alrubayyi et al. We have added the description of the control group and this additional analysis in Methods and Results of this revised version.

Table.

Unadjusted and adjusted mean difference at T1 values (log2 scale)				
	Unadjusted Mean (95% CI)	p-value	Adjusted* mean (95% CI)	p-value
Anti-RBD				
HCW	0		0	
HIV	-0.87 (-1.20, -0.53)	<.001	1.27 (-0.72, 3.27)	0.212
nAbs				
HCW	0		0	
HIV	0.04 (-0.43, 0.52)	0.853	1.24 (-1.37, 3.85)	0.353
IFN-gamma				
HCW	0		0	
HIV	-1.18 (-1.90, -0.46)	0.001	-4.98 (-8.52, -1.44)	0.006

*adjusted for value at T0, age, gender and time difference between T0 and T1 and between T-1 and T0

Importantly, the magnitude of specific T-cell responses was associated with the naïve CD4 T-cell pool size and the CD4/CD8 ratio in PLWH.

We have not investigated the CD4:CD8 ratio as an alternative stratification factor in this analysis as this was the research topic of another EU funded collaborative analysis which is merging our data with those of a number of other clinical infectious disease in Italy (the Vax-Icona Orchestra cohort). In such analysis the CD4:CD8 ratio was only weakly associated with the risk of having an undetectable anti-RBD response by one month after the second dose with SARS-CoV-2 vaccines (unpublished data).

Moreover, PLWH with CD4 T-cell below 200 cells/ μ L, a threshold for increased risk of AIDS, may not respond well and durably to vaccine, as previously reported for other vaccines.

This result is entirely consistent with our findings.

Vaccine response might, in any population, wanes due to developments of variants escaping T-cells, spillover and spillback of SARS-CoV-2 variants from human to animal then back and antibody selective immune pressure (i.e.: antigenic drift and to short-lived immune memory for SARS-CoV-2, which is not a virus with obligate viremic spread). Therefore, the today relevance for delta and omicron variants remains uncertain, limiting the clinical and public health relevance of the study findings.

We agree with this reviewer's comment and it is a limitation that we have acknowledged in the Discussion section.

Abstract and introduction:

AD additional dose is not clear, please replace by third dose (3D) for vaccine throughout the text and abstract

The acronym AD has been used in order to avoid misinterpretation between third additional (given to immune-suppressed subjects) and third booster dose (given to all subjects independently of immunologic status as a boosting dose after a well-established time period after the primary cycle). It has been modified according to your suggestion in third additional dose (AD) throughout the text and abstract.

Introduction: Introduction is too long to general as now several papers have been reported should immediately start with vaccine response in PLWH.

We have shortened the Introduction to reflect current guidelines for vaccination programs in PLWH.

These data are consistent with the observation that HIV infection may favor a poor serological response also to vaccines for viral agents, such as influenza. You need to add that low socioeconomic factors, professional exposition to COVID are also associated with PLWH, including co-morbidity that may be associated of more severe outcome, independently of immune status.

Fehr D,. Characterization of people living with HIV in a Montreal-based tertiary care center with COVID-19 during the first wave of the pandemic. AIDS Care. 2021 Mar 28:1-7

Indeed, we are aware that we cannot rule out unmeasured confounding bias. We have revised the Introduction and Discussion section by specifically adding socio-economic factors as a potential source of unmeasured confounding and by adding the aforementioned reference.

SID: <200/mm³; minor immunodeficiency, MID: 200-500/mm³; no immunodeficiency, NID: 171 >500/mm³. These abbreviations are confusing and it will be easier for readers to directly call groups less than 200, 200-500 and more than 500. It is incorrect to speculate that more than 500 equal no immunodeficiency in HIV. Change in the text and in table 1 and should be consistent with CD4 groups in Table 2.

We agree with the reviewer that previous abbreviations could be confusing. Considering that all our study population was on ART, it is more convenient for the aims of our analysis, to use a definition of CD4 strata based on the size of recovery. Therefore, we changed the terminology using the following acronyms: CD4 count <200 cell/mm³: poor CD4 recovery (PCDR); CD4 count between 200 and 500 cell/mm³: intermediate CD4 recovery (ICDR); CD4 count >500 cell/mm³: high CD4 recovery (HCDR). We have modified the abbreviations in the text, tables and figures. This nomenclature is also consistent with that used in other previous works from this group [Antinori A, et al. CID; 2022, accepted].

Contribution of CD4 CD8 ratio, should be presented as for some antibody vaccine response a better predictor than absolute CD4 counts.

We have not investigated the CD4:CD8 ratio as an alternative predictive factor in this analysis as this was the research topic of another EU funded collaborative analysis which is merging our data with those of a number of other clinical infectious disease in Italy (the Vax-Icna Orchestra cohort). In such analysis the CD4:CD8 ratio was only weakly associated with the risk of having an undetectable anti-RBD response by one month after the second dose with SARS-CoV-2 vaccines (unpublished data).

BNT162b2 (homologous P, HP); ii) primary series with mRNA-1273, third additional dose with mRNA-1273 (homologous M, HM); iii) primary series with BNT162b2, additional dose with mRNA-1273 (heterologous M, HtM); iv) primary series with mRNA-1273, additional dose with BNT162b2 (heterologous P, HtP). Strata specific mean values of RBD-binding IgG, nAb titres, IFN- γ with standard deviations were also shown. This mixture of vaccines sequences is hard to read and to keep in mind the numerous abbreviating proposed and then lower in the text. This part should be limited and presented as supplemental data, too difficult to follow in the current presentation.

The abbreviations have been simplified and reported in the main text; graphs presented as supplemental data.

Minor comments:

Reactogenicity should be reduced to a global commentary as the text is too long, no this chapter does not bring any new data. As reactogenicity is not generating new data, the term should be removed from the title and shortly discuss and presented data in a supplemental fig or table only.

We agree with the reviewer on this point. The term reactogenicity has been removed from the title. The findings have been shortened in the main text and reactogenicity according to vaccine sequence is shown only in supplementary figure 7.

Discussion is much too long and redundant with previous reports referred by investigators. Currently, several reports already focused on PLWH vaccine response. The discussion should mainly focus on PLWH with more than 350 and less than 200, which is the novelty of this paper. Throughout the text use the same term immunogenicity, humoral response, boosted immunity after initial course. Globally the text is not very clear and should be shorten with the focus on below CD4 T cell 200, which is the novelty here.

To our knowledge no new reports on response to a third booster dose in PLWH have appeared, although some additional studies on responses to the standard regimen have been published and a reference was added. (Jedicke et al. Humoral immune response following prime and boost BNT162b2 vaccination in people living with HIV on antiretroviral therapy. HIV Med. 2021 Nov 2:10.1111/hiv.13202. doi: 10.1111/hiv.13202. Online ahead of print. PMID: 34725907; Spinelli et al. Differences in Post-mRNA Vaccination SARS-CoV-2 IgG Concentrations and Surrogate Virus Neutralization Test Response by HIV Status and Type of Vaccine: a Matched Case-Control Observational Study. Clin Infect Dis. 2021 Dec 5:ciab1009. doi: 10.1093/cid/ciab1009. Online ahead of print. PMID: 3486496; Aledo et al. Safety and immunogenicity of SARS-CoV-2 mRNA-1273 and BNT162b2 vaccines in people living with HIV. AIDS. 2022 Jan 6). We have stressed the fact that the data on the comparison between CD4 count strata and vaccine sequence are completely novel.

Reviewer #2 (Remarks to the Author):

Vergori and colleagues investigated the safety and immunogenicity of an additional (third) vaccine dose for PLWH. The PLWH are further stratified based on CD4 count, which is used as a proxy for disease severity and efforts were undertaken to evaluate whether disease severity impacts on the response to the booster vaccine.

An important motivation for this study was the question whether PLWH, considered an immune-suppressed population, would benefit from a booster vaccine. This is definitely a justified question, but has lost some relevance as many governments by now have decided on providing booster vaccines to the entire adult population. Nonetheless, investigating whether whether the standard mRNA vaccination regimens are also effective for the population of PLWH and whether subsets of PLWH in terms of vaccine response exist remains a valid question.

Although the experiments reported in this study are overall well-performed, the study has several important limitations most of which are in fact identified by the authors in the discussion. Overall, this paper mainly confirms prior reports of good immune responses to SARS-CoV-2 vaccines in the global population of PLWH and shows that the 3rd dose, as in the general population, further enhances immunity in the majority of PLWH.

A major limitation is the fact that the study was underpowered for the ambitious aims of 1) identifying sub-populations of PLWH that would have suboptimal vaccine responses and 2) identifying optimal vaccination regimens for this subset.

We only partially agree with this comment. Although the analyses using binary endpoints were clearly underpowered, this is not the case for those using the continuous endpoint for which the power is much larger by definition. Indeed, the ANOVA analysis carry strong evidence against the null hypothesis of no difference in humoral response between ICDR vs. HCDR (mean \log_2 0.5, std error 0.14, $p=0.002$) (supplementary Figure 2 A-C). Similarly, the data carried strong evidence for a difference in mean \log_2 increase of anti-RBD IgG comparing T_0 to T_1 according to vaccine sequence with heterologous being superior to homologue vaccination, especially when the boosted dose was by Moderna mRNA-1273 (supplementary Figure 6 A). We have now revised the Discussion section to highlight these key findings from the association analyses so that they are no longer buried in the manuscript or only shown as Supplementary material.

Also, the fact that the study did not contain a healthy control group makes it difficult to assess the extent to which the response of the PLWH cohort to the AD differs from that of the general population. Taken together, although several interesting trends are reported, drawing firm conclusions from this study is difficult.

We have now performed an additional analysis of covariance analysis with the aim to compare humoral and SARS-CoV-2 T-cell specific between our cohort of PLWH and a control group of HIV negative health care workers (HCWs) after controlling for gender, age and difference in the exact days interval between the booster dose and the measured response. PLWH indeed showed impaired humoral and T-cell mediated responses than those seen in matched HCW. See response to Reviewer #1 to this point and new results shown in the paper.

Specific comments

- Abstract line 36: “we analyzed anti-RBD, microneutralization assay and IFN γ production in 216 PLWH on ART with advanced disease (CD4 count <200 cell/mm 3 and/or previous AIDS) at the time of their first dose”: it is not clear to me how this population is then stratified at T_0 in the severe immunodeficiency (SID: <200/mm 3); minor immunodeficiency (MID: 200-500/mm 3) and no immunodeficiency, (NID: >500/mm 3) groups that are used for statistical analyses. What are the PLWH with advanced disease when they received their first vaccine dose that end up in the MID and NID groups at T_0 ? Are the CD4 counts that variable between the first vaccine dose and the AD? In that case this does not seem like a very good parameter to use for stratification.

This is correct. Although the target population includes participants who had a CD4 count <200 cell/mm 3 and/or previous AIDS at the time of their first dose (baseline) by the time T_0 (a median of xx days after baseline) CD4 count might have increased in some so that they distributed as CD4 count <200 cell/mm 3 (n=44); CD4 count between 200 and 500 cell/mm 3 (n=96) and CD4 count >500 cell/mm 3 (n=76). This is the level of immunosuppression measured at the time of receiving the AD which is the predictor of interest in our analysis (current level of immunosuppression).

From Table 2 it does seem that the MID and NID groups are enriched for PLWH with prior AIDS. Is it correct to assume then that these have higher CD4 counts? That seems a bit counterintuitive

This is also correct. Participants with CD4 count >200 cell/mm³ were eligible for the vaccination program at baseline only if they had been previously diagnosed with AIDS. This explains the higher proportion of patients with AIDS in the MID and NID groups. Of note, because these are patients receiving ART and we were interested in the current level of recovery on ART we have modified the definition of CD4 strata based on the size of recovery using the following acronyms: CD4 count <200 cell/mm³: poor CD4 recovery (PCDR instead of SID); CD4 count between 200 and 500 cell/mm³: intermediate CD4 recovery (ICDR instead of MID); CD4 count >500 cell/mm³: high CD4 recovery (HCDR instead of NID).

Are the CD4 counts at time of first dose known for the patients with prior AIDS? Perhaps a figure with the evolution of CD4 counts from first vaccine dose to T0 can be included for all 216 participants?

The table below shows the transition in CD4 count from baseline to T0. CD4 count at baseline was available for all participants. Thus, 26% of those who had a CD4 count of 0-200 cell/mm³ at baseline, moved up to a level higher than PCDR. In contrast, only 2/44 (5%) of participants who were classified as PCDR at T0 had started the vaccination program with a baseline CD4 count >200 cell/mm³. We have added this table as Supplementary material.

Baseline CD4 (cell/mm ³)	T0 group			Total
	PCDR	ICDR	HCDR	
0-200	42 (74%)	15 (26%)	0 (15%)	57 (100%)
201-500	2 (2%)	74 (81%)	15 (16%)	91 (100%)
500+	0 (0%)	7 (10%)	61 (90%)	68 (100%)
Total	44	96	76	216

- Line 153: “The detection limit of these assays was 0.17 pg/mL for IFN-γ and the cut off used in this analysis to define the T specific cells response was 12 pg/mL.” What was the rationale for selecting this cut-off? Likewise: how were the cut-offs for other assays determined? For the Nab dilution it is stated that the cut-off is 1:128 (line 253). Is this correct? Some points in the graphs have a log2 dilution over 10 (hence dilution of over 1000?). In the mat & meth section (line 138) it is stated that “7 two-fold serial dilutions (starting dilution 1:10)” was used which would indicate that the highest dilution tested would be 1:640 if I am correct? Can this be clarified?

The cut-off of a positive T cell response (12 pg/ml) has been calculated as the mean + 2SD of the response to spike peptides of unvaccinated uninfected healthy donors [Agrati C, Castilletti C, Goletti D, et al, On Behalf Of The INMI Covid-Vaccine Study Group. Coordinate Induction of Humoral and Spike Specific T-Cell Response in a Cohort of Italian Health Care Workers Receiving BNT162b2 mRNA Vaccine. Microorganisms. 2021 Jun 16;9(6):1315. doi: 10.3390/microorganisms9061315. PMID: 34208751; PMCID: PMC8235087]. This definition has also been added to the methods section with reference.

The cut off 1:128 was a mistake, the inaccuracy was corrected in the manuscript. The last dilution was 1:640 but some of the samples showed a titer \geq 1:640, to these samples was assigned an arbitrary title of 1:1280. A sentence was added in materials and method section.

- It would be helpful to show the cut-offs used for qualitative statistical analyses on the graphs. The cut off values are now reported in each graphs.

- Fig. 1-3: the use of “increase” on the title is confusing as this suggest a delta value is represented which is not the case and moreover some individuals show a decrease. Also, for Fig. 1 the use of “peak” in the title is inappropriate as there are no longitudinal data showing that the sampling time point indeed represents the peak response.

We have now reworded the titles of the aforementioned figures by replacing ‘increase’ with ‘change’.

We have also reworded ‘peak’ with ‘1 month response post second dose’.

- Line 181: “In the ANOVA analysis we used the naïve approach of replacing the truncated values with the upper limit of the assay for anti-RBD >7.1 BAU/mL and nAbs $>1:10$ ”. Can you clarify why this approach was chosen.

It is a method often used in HIV research for censored HIV-RNA values and often called the ‘crude’ or ‘naïve’ method [Marschner IC, et al. Use of changes in plasma levels of human immunodeficiency virus type 1 RNA to assess the clinical benefit of antiretroviral therapy. J Infect Dis. 1998 Jan;177(1):40-70]. It is a crude method that allows inclusion of all participants so that standard statistical procedures (such as ANOVA) can be performed but may introduce bias if the proportion of participants with a truncated value is high [Flandre P et al. On the use of magnitude of reduction in HIV-1 RNA in clinical trials: statistical analysis and potential biases. J Acquir Immune Defic Syndr. 2002 May 1;30(1):59-64]. For this reason we also performed the truncated linear regression model analysis.

- Line 189: “truncated regression analysis was similar to ANOVA but was fitted on the natural scale of the responses and correctly accounted for truncated values (participants who reached the upper limit of the assay for specific responses).” Do you mean that no log transformation was done here?

This is correct. This is a semi-parametric approach so that the response variable does not need to be normally distributed and the assumption that error variance is constant across observations can also be relaxed. However, we have repeated the analysis using the log-transformed values and the results were similar with overall Wald test p-value <0.0001 to reject the null hypothesis of no difference in anti-RBD IgG levels by both the CDR and vaccine sequence groups.

- Line 237: There is an association between IgG titers and CD4 counts but not between neutralization capacity and CD4 counts. Do IgG and neutralization titers correlate?

We did find an association between neutralization capacity and CD4 counts strata although the analysis was likely to be underpowered [aOR 7.45 for PCDR vs. HCDR (95%CI 0.33-168.1); p=0.21]. As expected, anti-RBD and nAbs at T1 were highly correlated (see Figure below, Pearson correlation coefficient $\rho=0.86$, $p<0.0001$).

- The discussion focuses on comparison to other immune deficient populations such as individuals taking rituximab or chemotherapy. This might not be the most relevant comparison. It would be more interested to discuss what is known on the immune dysfunction that is characteristic for PLWH and how this impacts vaccination.

The discussion has been modified according to this suggestion and comparisons to other immune-suppressed patients was partially shortened.

- The authors state in the discussion that “The achieved responses are particularly impressive, considering the participants’ status of chronic immune activation, as well as the fact that depletion of viral-specific T and B cell clones is observed even in PLWH responder to antiretroviral therapy (cART)”. Is there information on the inflammatory status of the study participants?

The term immune activation was a typo, it has been corrected and replaced by immune ‘dysregulation’. No information on the inflammatory status are available.

Minor

- The manuscript still contains quite a bit of grammatical errors: use of articles, singular-plural mistakes. Also, some sentences are formulated in a way that makes them difficult to grasp the message. Perhaps an additional round of language editing would be advisable.

We have now performed an additional round of language editing as suggested.

- Although this is clearly described in the Methods, from the abstract it is difficult to understand what is meant by the “additional dose” (AD). Perhaps it can already be clearly stated in the abstract that the additional dose is the third vaccination given to individuals that have completed the standard two-dose mRNA vaccination regimen (either Pfizer or Moderna). And that the investigated population includes both recipients of a homologous and heterologous AD.

We agree with the reviewer on this point and we have now clearly stated the definition of the additional third dose in the abstract. Details regarding the type and sequence of vaccine received have been instead omitted in the abstract due to the number of words limit.

- Although the authors are right that no reports on response to third doses in PLWH are available, some additional studies on responses of PLWH to the standard regimen could be referenced (eg. Jedicke et al. Humoral immune response following prime and boost BNT162b2 vaccination in people living with HIV on antiretroviral therapy. HIV Med. 2021 Nov 2:10.1111/hiv.13202. doi: 10.1111/hiv.13202. Online ahead of print. PMID: 34725907; Spinelli et al. Differences in Post-mRNA Vaccination SARS-CoV-2 IgG Concentrations and Surrogate Virus Neutralization Test Response by HIV Status and Type of Vaccine: a Matched Case-Control Observational Study. Clin Infect Dis. 2021 Dec 5:ciab1009. doi: 10.1093/cid/ciab1009. Online ahead of print. PMID: 3486496; Aledo et al. Safety and immunogenicity of SARS-CoV-2 mRNA-1273 and BNT162b2 vaccines in people living with HIV. AIDS. 2022 Jan 6. doi: 10.1097/QAD.0000000000003161. Online ahead of print. PMID: 34999608)

We thank you the reviewer for the suggestion. We have now added and made appropriate references to these articles.

- Line 53: nevertheless? Seems like “in addition” or something alike would be better suited here?

Right, we have replaced ‘nevertheless’ with ‘in addition’.

- Line 59: “the wild-type D614G virus”; perhaps rename as G614 SARS-CoV-2 variant

The sentence containing this term has been removed following the suggestion of reviewer #1.

- Line 67: “Several studies recently supported a significant increase of neutralization against Omicron after a booster dose, even lower than that observed with ancestral type or Delta suggesting a cross-reactivity of neutralizing antibody responses”: do you mean that the increase in neutralization is lower?

Yes. The sentence has been reworded as below

“Several studies recently supported a significant increase of neutralization against Omicron after a booster dose, although this increase was lower than that observed with ancestral type or Delta VoC suggesting cross-reactivity of neutralizing antibody responses”.

- Line 114: “consecutive PLWH”: what do you mean by this?

Consecutive sampling was used to enroll PLWH in the study. We have reworded the sentence accordingly.

- Line 116: “Individuals with a SARS-CoV-2 infection diagnosis, defined by a RT-PCR positive to the molecular test on the nasopharyngeal swab, or positivity to anti-N IgG, were excluded for the present analysis.” Were these analyses conducted on samples from T0 and T1?

Participants were tested for anti-N IgG only at T0 and excluded from the analysis if they showed a positive result. We have reworded the sentence to better clarify this point.

- Line 123: “In addition, both humoral neutralizing antibodies and T-cell responses measured approximately 30 days after participants’ second vaccine dose were also available for comparative analysis (time T-1).” Is the reason that these were not available for all participants that a major subset had a smaller gap between dose 2 and dose 3 (AD)?

It is possible that some participants in HIV-VAC were still not eligible for the 3AD because at the time in which the database was frozen less than 28 days had elapsed from the date of receiving the second dose. These have been excluded by default from all analyses. None of the participants who received the AD have been excluded on the basis of the time from the date of the second dose. Elapsed time was fairly variable as shown in Table 1 and we have now controlled the analysis for this additional potential confounder. Simply 47 participants (22%) did not return for the 3AD dose after the second dose despite being eligible and are excluded from the T-1 analysis.

- Line 226: “The humoral response elicited by AD was on average stronger than the peak titres elicited by the primary vaccination 1 month after the completion of 2 doses vaccination cycle” : is it appropriate to say that the response to the AD is stronger as different timepoints post-vaccination are compared?

We agree with the referee that the time point after the first vaccination cycle and after the AD was different, weakening the comparison. Nevertheless, a very effective response to the third dose was reported and should be highlighted. We modified the text according to this suggestion.

- Line 336: Are nasal congestion and heavy breathing not unusual vaccine side effects? Where these reported in the recent study by Munro et al. that is also cited in the discussion?

The sentence is referred to the symptoms fatigue, headache, myalgia and fever. We agree with the reviewer that nasal congestion and heavy breathing are not usual for vaccination (and not reported in the paper by Munro et al), nonetheless these were reported by patients during the telephone interview.

- Line 393: "The T cell contraction observed overtime can be effectively reverted by the additional dose, with some efficacy as reported by the primary vaccination schedule": What do you mean by this?

The sentence has been reworded as follows: The T cell response that contracted overtime after the first two doses can be effectively boosted by the additional dose.

- Table 1: the title (current CD4 count) is confusing and not necessary. The readability of the table could be improved by using a format where table cells have borders.

The Table 1 has been reformatted as suggested. However, we believe that it is important to specify that the analysis is stratified according to CD4 count measured at T0 (current value) rather than those at other time points.

- Table 2: why not replace CD4 T cell counts by SID, MID, NID?

The table 2 has been modified using the new acronyms for the strata.

- Line 452: "These prevalences...": which prevalences

The word 'prevalences' has been replaced with 'proportions'.

Reviewer #3 (Remarks to the Author):

Vegori et al characterize humoral and cellular responses to a third dose of COVID-19 mRNA vaccine in individuals living with HIV who had met the definition of advanced disease (CD4 count <200 cells/mm³ or prior AIDS diagnosis) at the time of their first COVID-19 vaccine dose. The team also characterize reactogenicity. The results are timely, as there is presently little information on immune responses to 3rd COVID-19 vaccine doses in people with HIV. The cohort size (N=216), the fact that there are two historic longitudinal datapoints following the 2nd dose for most participants, and the fact that the researchers measured both humoral and cellular responses, are additional strengths of the study.

However, I have a number of major concerns, many that relate to unclear descriptions/justifications of the statistical analyses performed, that substantially dampen my enthusiasm for the manuscript.

1. Data are presented in a redundant manner. For example, the immune response data from the T1 timepoint are presented in Figure 1 (where these values are compared to T-1), Figure 2 (where these values are compared to T0), Figure 3 (same data as Figure 2 but stratified by CD4 count) and Figure 4 (essentially same data as Figure 2 but stratified by vaccine regimen),

Supplemental Figures 2-4 (same data as Figure 1 but stratified by CD4 count), with separate multivariable analyses adjusting for CD4 count and vaccine regimen, respectively, presented in Tables 2 and 3.

The authors should instead clearly delineate the paper's main goal(s)- e.g. to characterize the relationship between CD4 count and post-3rd dose responses in PLWH - and present these data in a succinct set of figures, with a single multivariable model that adjusts for other possible confounders including vaccine regimen.

Values at T1 were included in separate Figures because the number of participants in the various analyses is not the same. Indeed, the value at time T-1 (one month after the second dose) was available only for a subset of the participants included in the main analysis comparing values at T0 and T1. Similarly, we had two separate exposures of interest (CD4 count at T0 and vaccine sequence) and we evaluated these associations in separate models (hence the need for a separate Figure/Table is needed). In fact, the set of identified confounders is different in the model with exposure CD4 count vs the model with exposure vaccine sequence. Finally, we agree that the Figures showing the overall changes over time in responses are partially redundant but the spaghetti plot shows additional data details (i.e. how many individual patients showed an increase or a decrease over time for the various responses) not included in the analysis showing the mean population change.

2. Statistical analyses are incompletely described and justified. Some examples are:

a) Use of a binary cutoff set at each humoral assay's lower limit of quantification (LLOQ) is not appropriate for a post-3rd dose vaccine immunogenicity study as the number of non-responders is essentially zero (e.g. there are only 3 non-responders out of all samples shown in Figure S2). If a binary cutoff remains appropriate for the cellular immune analyses, the value (12 pg/mL) should be justified.

We believe that the proportion of non-responders is important to show. Although, it is a very small number these are patients at risk of severe disease if they acquire the infection and we are in the process to further characterize them in a separate study. We agree that the associations analyses are of limited values and this is why we also compared mean titres values.

The cut-off of a positive T cell response (12 pg/ml) has been calculated as the mean + 2SD of the response to spike peptides of unvaccinated uninfected healthy donors [Agrati C, Castilletti C, Goletti D, et al, On Behalf Of The INMI Covid-Vaccine Study Group. Coordinate Induction of Humoral and Spike Specific T-Cell Response in a Cohort of Italian Health Care Workers Receiving BNT162b2 mRNA Vaccine. Microorganisms. 2021 Jun 16;9(6):1315. doi: 10.3390/microorganisms9061315. PMID: 34208751; PMCID: PMC8235087].

b) lines 193-197. How did the authors select the variables to include in the DAG analysis, and why was this approach chosen? Table 1 for example suggests additional variables that could serve as confounders (e.g. liver disease, dose timing), why were these not included?

The DAG was constructed on the basis of causal links established in randomized studies or other axiomatic knowledge. Indeed, time from the second dose appears to be a common cause of both CD4 count at T0 and vaccine response and therefore likely to introduce confounding. In contrast, while liver disease may have an impact on CD4 count (especially in patients with replicating HCV-RNA) we cannot find a biological or probabilistic reason by which it should affect vaccine responses, apart from people with cirrhosis [Ai J, Wang J, et al. Safety and Immunogenicity of SARS-CoV-2 Vaccines in Patients With Chronic Liver Diseases (CHESS-NMCID 2101): A Multicenter Study. Clin Gastroenterol Hepatol. 2021 Dec 20:S1542-3565(21)01346-X. doi: 10.1016/j.cgh.2021.12.022. Epub ahead of print. PMID: 34942370; PMCID: PMC8686447]. Only 4 participants of those who were classified as having liver disease were actually cirrhotic (severe liver disease, 2% of study population) and therefore liver disease is an instrument in our analysis and would only introduce noise for our estimates if included in the models. We have revised the DAG and the models with CD4 at T0 as the main exposure of interest accordingly.

Related to this, what factors were included in the multivariable models presented? Lines 198-199 indicate that only age adjustment was performed, but the text (e.g. lines 292-296) lists additional variables for certain analyses, though not others.

Our assumptions regarding the causal links in the data are described transparently in the DAG. For the model with CD4 count as the main exposure of interest we identified a set of potential confounders: age, time from HIV diagnosis, CD4 nadir, cancer and HIV-RNA at the time of 3AD vaccination (lines 188-189). To these we followed your suggestions and have now added the time from the date of receiving the second dose. In contrast, vaccine-sequence was pseudo-random so the model for this exposure was only adjusted for age.

c) The comparisons performed in Figures S2-S4 are not clear. How were these data reduced to frequency summaries needed to apply Fisher's exact or chi-squared tests?

The F test refers to the global F test to determine whether there is evidence to reject the null hypothesis of equality of the means according to CD4 count recovery groups, not the Fisher exact test. Thus, for example, Figure 2A with a Fisher test p -value < 0.0001 indicates that there is sufficient evidence to reject the null hypothesis that all means \log_2 increase of anti-RBD IgG from T-1 to T1 are the same in the groups.

What data were compared to generate the p -value in the title of each figure, and how does this differ from the comparisons that were performed to generate the p -values above the figure panels?

After having established that there was at least one difference in the means we proceeded to test the specific contrasts (HCDR vs. PCDR; HCDR vs. ICDR and ICDR vs. PCDR) using Bonferroni corrected p -values. These p -values for the contrasts are shown above the figure panels. Thus, again picking Figure 2A as the example, this means that the Fisher p -value was driven by the specific contrast HCDR vs. ICDR. We have added a footnote to indicate that the p -values above the figure panels are Bonferroni-corrected values.

Related to this, in Figure S3, how can the p-value in the title be highly significant when none of the other comparisons are significant? The same issue affects Figure 3.

It is indeed possible because the F test does not correctly takes into account multiple comparisons and the inflation of type I error. We have added a footnote to indicate that the p-values above the figure panels are Bonferroni-corrected values.

As mentioned above, binary cutoffs are not appropriate for the humoral data anyway. But, part of the confusion here may be that the figure highlights a change in response between two timepoints, presented on a continuous scale, when the authors may in fact be converting results from only one of these two timepoints to a binary outcome, and comparing those frequencies in 2x2 tables (?)

No, this is not the corrected interpretation of what we have done (see responses above).

d) lines 275, 282, 296: on both these lines, the authors state that a multivariable ANOVA was performed, and then report a p-value for a Fisher's exact test. Are the authors instead referring to the model F-statistic?

Yes, this is a typo. We are referring to the global F test to determine whether there is evidence to reject the null hypothesis of equality of the means. The test was derived by the same Sir Ronald Fisher who developed the exact test.

Moreover, when describing multivariable analyses, it is essential to clearly state what outcome is being evaluated, as well as what variables were included in each model. For example, in lines 279-285, the text suggests that the outcome variable is the change between T0-T1 (though this should read T1-T0), but the figure suggests that the outcome variable was the magnitude of the response at T1.

We agree with this point. The outcome variable was the change T1-T0. We have modified the text in the Figure accordingly.

Furthermore, the text indicates that the model adjusted for age, but what about other potential confounders?

We could not identify any other factor besides perhaps age that could have affected the choice of vaccine sequence which was determine pseudo-randomly. Thus, by definition there are no other measured confounders that could have biased the association of interest. Of course we cannot rule out unmeasured confounding which we have acknowledged in the Discussion.

e) The same question of what data are being compared affects Figures 4A-C. Here, are authors comparing the mean increase between timepoints (as indicated in the figure title) or the responses at T1 (as seems to be indicated in the figure)

The outcome variables in all ANOVA models was the change T1-T0. We have modified the text in all Figures accordingly.

f) The authors should use caution if and when analyzing any outcome variable that represents the magnitude of change between a prior timepoint and T1, as a substantial minority of values at T1 (particularly for anti S-RBD IgG) are above the assay limit of quantification, which means that any magnitudes of change for these participants will be underestimates of the true change. As the latter occurs most often in the NID group, this could lead to bias.

We agree with this point. This is the reason why we performed the additional truncated linear regression model which correctly accounts for censored data.

g) lines 189-190 (truncated regression analysis) are unclear: what do the authors mean when they say that "ANOVA was fitted on the natural scale of the responses and correctly accounted for truncated values?"

The full sentence read as follows: "The truncated regression analysis was similar to ANOVA but was fitted on the natural scale of the responses and correctly accounted for truncated values (participants who reached the upper limit of the assay for specific responses)."

It is the truncated linear regression analysis that correctly accounts for censored data not the ANOVA which used the crude method of imputing censored values with the upper limit of detection. We have now repeated the truncated linear regression analysis using values in the \log_2 scale and results were similar.

3. Some comments in the discussion appear to contradict the data presented, though this could be due to a lack of clarity, again on what outcome is being discussed. For example:

- a) line 351 states that "no significant association could be found with... CD4 count" but this contradicts lines 237-239 "we found an association between the level of CD4 count at T0 and the observed variations in humoral responses from ...[T-1 to T1]" and lines 414-416 "our data suggest that the CD4 count at time of receiving the AD is a potential determinant of the magnitude of the response...."

We agree that the sentence was confusing and we have reworded it as follows:

"Although overall the data carried little evidence for an association between current level of CD4 count and response to vaccine, we found a significant difference in humoral response when comparing participants with poor CD4 count recovery on ART (PCDR) with those high CD4 count recovery on ART (HCDR). Also for the other responses there was a tendency for the (PCDR) to show worse outcomes than (HCDR), although results were not significant likely because of low statistical power."

b) The authors state on lines 422-423: "participants who received as primary series BNT162b2 followed by an AD with mRNA-1273 showed a more robust humoral response than those received other sequences". But, in Figure 4, the highest T1 responses were observed in people who had received three mRNA-1273 doses. It is possible that the authors are referring to the CHANGE in responses between T0/T1, but this is not clear (and please see caveat in comment 2f).

This is correct, the outcome of interest is the change T1-T0 which was the greatest for the HtM group (in blue in Figure 4). We have corrected the text and figures accordingly.

4. The paper could be additionally strengthened with the inclusion of data from a control group without HIV - it appears from the author's prior work that these participants were recruited

We performed an additional analysis of covariance analysis with the aim to compare humoral and SARS-CoV-2 T-cell specific between our cohort of PLWH and a control group of HIV negative health care workers (HCWs) after controlling for gender, age and difference in the exact days interval between the booster dose and the measured response. PLWH indeed showed impaired humoral and T-cell mediated responses than those seen in matched HCW. See response to Review#1 and revised paper to see the results of this analysis.

5. Given the rapid spread of Omicron worldwide, the paper could be additionally strengthened with Omicron-specific assays.

We agree with the reviewer, these tests will be performed in the near future. The study covers a period in which alpha/delta variants prevailed as documented by the test performed at this times.

Additional comments:

1. Vaccine regimen type should be added to Table 1

Good point. We have now added the breakdown of vaccine sequence in Table 1.

2. On all Figures, consider adding the assay lower and upper limits of quantification. If binary cutoffs are retained, consider adding these as well.

A line for cut off values have been added in the Figures.

3. For univariable analyses, consider reporting median/IQR rather than mean and applying nonparametric statistics, as some of the data appear non-normal even after transformation. Also, put p-values on each panel, not in the figure title.

We had carefully tested for the normality assumption before performing the parametric tests. We have moved the p-values inside the Figures as requested.

Reviewers' Comments:

Reviewer #1:

Remarks to the Author:

Issues have been addressed

Study findings are novel and timely

Reviewer #2:

Remarks to the Author:

The authors have adequately addressed my comments. I have no further questions.

Reviewer #3:

Remarks to the Author:

Vergori et al have modified their manuscript to include a control group of people without HIV (n=98) and have provided responses to concerns raised by the reviewers. Both reviewers 2 and 3 had raised concerns regarding the statistical analyses. In my opinion, the revisions do not adequately address these concerns.

A manuscript investigating post-3rd dose COVID-19 vaccine responses in PLWH should clearly answer two main questions:

1. Following 3-dose COVID-19 vaccination, and after controlling for relevant sociodemographic, health and vaccine-related variables (e.g. regimen type, dose interval), do the magnitudes of humoral and cellular immune responses differ between PLWH and controls?
2. Following 3-dose COVID-19 vaccination, and after controlling for relevant sociodemographic, health and vaccine-related variables (e.g. regimen type, dose interval), do the magnitudes of humoral and cellular immune responses in PLWH differ by the most recent CD4+ T-cell count?

In my opinion the revised manuscript still does not clearly address these questions, for the following reasons:

-Though a control group of people without HIV was added, no data from this group was incorporated into any figure or Table, just summarized briefly in the text. This information is very important however, as it appears that cellular (though not humoral) responses were weaker in PLWH compared to controls.

- The authors still do not adjust for vaccine-related variables (including regimen type and dose interval) in their primary multivariable analyses of the relationship between HIV (and CD4 T-cell count in PLWH) and COVID-19 vaccine responses. The authors justify this by stating that they are interested in exploring the effects of vaccine regimen type separately from the effects of HIV/CD4+ T-cell count. But, this is not appropriate for a paper whose major goal is to investigate the effects of HIV (and CD4+ T-cell count) on COVID-19 vaccine responses. Controlling for vaccine regimen is especially relevant given the authors' discovery that vaccine regimen type influenced vaccine responses, and that the three CD4 strata in PLWH differed in terms of their vaccine regimen type (as reported in table 1; $p=0.02$).

- I maintain that it is not appropriate to use a binary classifier (responder/nonresponder) as a primary analysis when the number of non-responders is essentially zero (as one would expect following three

vaccine doses). While the authors can (and should) report rare occurrences of non-response, my point was that the primary statistical analyses should not be based on these. The only multivariable analyses presented in the paper are Tables 2 and 3, which use binary outcome classifiers.

- I acknowledge that the authors do perform multivariable analyses on the quantitative values as secondary analyses, in the form of a "truncated regression analysis". The description of this analysis however, which both reviewers 2 and 3 indicated was unclear, remains unchanged in the revised manuscript (lines 190-193).

Additional comments:

1. The use of the terminology "third additional dose" is confusing. Reviewer 1 had raised this point, and had suggested the authors use "third dose".

2. There are still errors and typos throughout. For example, the Ns reported in the text for vaccine regimen types (lines 232-235) only add up to 212, not the reported total n=216. Moreover, these numbers do not match those in Table 1, which only add up to 214.

Rome, 18th May 2022

We here provide a point-by-point reply to the comments and we have incorporated the related changes in the manuscript.

We thank the rev#3 for his thoughtful insights, which helped to significantly improve the manuscript.

Reviewer #3

Reviewer #3 (Remarks to the Author):

Vergori et al have modified their manuscript to include a control group of people without HIV (n=98) and have provided responses to concerns raised by the reviewers. Both reviewers 2 and 3 had raised concerns regarding the statistical analyses. In my opinion, the revisions do not adequately address these concerns.

A manuscript investigating post-3rd dose COVID-19 vaccine responses in PLWH should clearly answer two main questions:

1. Following 3-dose COVID-19 vaccination, and after controlling for relevant sociodemographic, health and vaccine-related variables (e.g. regimen type, dose interval), do the magnitudes of humoral and cellular immune responses differ between PLWH and controls?

Our data show that PLWH, after controlling for the potential confounders mentioned, did not show significantly different anti-RBD and nAbs responses compared to the control group of HCW. In contrast, the data carried significant evidence for a lower mean difference in IFN-gamma between PLWH and controls. Thus, these are important results as they show that despite a significant response to a 3rd dose, PLWH appear to have an impaired T-cell mediated response as compared to the general population.

We have now included more details of the analysis comparing average levels of responses in this second revised version. We have now made the results of this analysis a lot more prominent in the main text, including a new figure comparing mean titres between PLWH and the HCW. Mean values and difference by groups from fitting linear regression models are shown in the table below and the new Figure 3a and the revised text (lines 368-370, 383-385 and 444-446). Our data are consistent with those shown by others [e.g. Alrubayyi et al.]

Table 1. Naïve unadjusted means from fitting the ANOVA model

Marker (log2 scale)	PLWH	HCW	F test p-value
RBD-binding IgG, mean (SD)			0.21
T0	7.3 (2.8)	6.1 (1.1)	
T1	11.7 (2.3)	12.0 (0.9)	
T1-T0	4.5 (1.9)	5.9 (1.2)	
nAb titres			0.35
T0	5.0 (2.1)	3.4 (1.4)	
T1	8.7 (2.1)	8.0 (1.3)	
T1-T0	3.7 (2.2)	4.6 (1.7)	
IFN- γ			0.006
T0	6.7 (3.5)	7.7 (1.8)	
T1	7.4 (2.9)	8.7 (1.7)	
T1-T0	0.8 (3.0)	1.7 (1.5)	

Table 2. Adjusted difference in means from fitting an ANCOVA model (contrasts between groups)

	Unadjusted and adjusted mean difference at T1 in anti-RBD (log2 scale) HIV vs. HCW			
	Unadjusted Mean (95% CI)	p-value	Adjusted* mean (95% CI)	p-value
RBD-binding IgG	-0.87 (-1.20, -0.53)	<.001	1.27 (-0.72, 3.27)	0.212
nAb titres	0.04 (-0.43, 0.52)	0.853	1.24 (-1.37, 3.85)	0.353
IFN- γ	-1.18 (-1.90, -0.46)	0.001	-4.98 (-8.52, -1.44)	0.006

*adjusted for value at T0, age, gender and time difference between T0 and T1 and between T-1 and T0

2. Following 3-dose COVID-19 vaccination, and after controlling for relevant sociodemographic, health and vaccine-related variables (e.g. regimen type, dose interval), do the magnitudes of humoral and cellular immune responses in PLWH differ by the most recent CD4+ T-cell count?

We agree that results of the analysis of the association with current CD4 count were not controlled for vaccination sequence and time from last dose of vaccine received. Indeed, according to our assumptions described by means of a DAG (see Figure), time since last vaccination dose was a confounder of the association (pink node) so the lack of its inclusion in the model was an oversight, while the type of vaccination was only a predictor of outcome (because it is unlikely that vaccine sequence has an effect on current CD4 count).

However, including predictors of outcome in a regression model increase the efficiency of the estimates and therefore we have now included both these factors in all models.

Results are shown below. Overall, the new results carry little evidence for an association between current level of CD4 count and response to vaccine. There was a signal for participants in the PCDR group to have greater risk of remaining undetectable for neutralizing antibodies and T-cell mediated responses after the 3rd dose as compared to participants in the HCDR but differences were no longer significant when controlling for key confounding factors. Results from comparing average levels in the naive ANCOVA analysis were largely inconclusive showing little differences between the groups. Even the truncated linear regression analysis, carried no evidence for an overall effect of CD4 count in RBD-binding IgG ($p=0.20$) and neutralizing antibodies response ($p=0.26$). New results are shown in the Tables below. We have revised the text by including the mean responses in the groups and the corresponding paragraph of the Conclusions according to these new data.

Table 1. Main unadjusted mean differences from fitting the ANOVA model

Marker (log2 scale)	Current CD4 count			F test p-value
	PCDR 0-200	ICDR 201-500	HCDR 501+	
RBD-binding IgG, mean (SD)				0.19
T0	5.1 (3.7)	7.4 (2.4)	8.4 (1.7)	
T1	10.2 (3.7)	12.1 (1.5)	12.3 (1.2)	
T1-T0	5.1 (2.2)	4.7 (2.1)	4.0 (1.4)	
nAb titres				0.18
T0	3.7 (2.2)	5.1 (2.1)	5.7 (1.8)	
T1	7.3 (2.9)	9.0 (1.9)	9.1 (1.5)	
T1-T0	3.6 (2.4)	4.0 (2.3)	3.5 (2.0)	
IFN- γ				0.87
T0	4.0 (3.6)	7.0 (3.2)	7.4 (3.3)	
T1	4.7 (3.7)	7.7 (2.2)	8.5 (2.2)	
T1-T0	0.6 (1.7)	0.6 (3.1)	1.1 (3.3)	

Table 2. Adjusted mean differences from fitting an ANCOVA model (contrasts between groups)

Unadjusted and adjusted mean difference at T1 (log2 scale)				
	Unadjusted Mean (95% CI)	p-value	Adjusted* mean (95% CI)	p-value
RBD-binding IgG				
501+	0		0	
201-500	0.66 (0.09, 1.24)	0.025	0.50 (-0.08, 1.08)	0.093
0-200	1.09 (0.38, 1.80)	0.003	0.05 (-0.84, 0.94)	0.910
nAb titres				
501+	0		0	
201-500	0.51 (-0.16, 1.17)	0.136	0.27 (-0.47, 1.01)	0.478
0-200	0.13 (-0.69, 0.95)	0.755	-0.70 (-1.84, 0.45)	0.234
IFN-gamma				
501+	0		0	
201-500	-0.43 (-1.36, 0.49)	0.358	-0.03 (-1.15, 1.10)	0.962
0-200	-0.43 (-1.59, 0.73)	0.470	-0.43 (-2.18, 1.31)	0.628

*adjusted for age, time from HIV diagnosis, CD4 count nadir, VL<=50 copies/mL at T0, days from the date of 2nd dose, vaccine sequence, and concomitant cancer

&from the adjusted model

Table 3. Logistic regression analysis

	Logistic regression of the risk of undetectability post booster				&Type III p-value
	Unadjusted		Adjusted*		
	Odds ratio (95% CI)	p-value	Odds ratio (95% CI)	p-value	
CD4 count (cells/mm³)					
	nAbs				
501+	1		1		0.549
201-500	1.61 (0.14, 18.13)	0.699	1.49 (0.08, 27.99)	0.791	
0-200	11.84 (1.38, 101.9)	0.024	5.04 (0.22, 115.1)	0.311	
	IFN-gamma				
501+	1		1		0.392
201-500	1.59 (0.28, 8.94)	0.598	0.61 (0.07, 5.54)	0.657	
0-200	15.00 (3.15, 71.37)	<.001	2.48 (0.29, 21.56)	0.410	
	nAbs at T6=1280				
501+					0.276
201-500	1.37 (0.75, 2.52)	0.304	1.33 (0.62, 2.84)	0.464	
0-200	0.52 (0.24, 1.12)	0.093	0.58 (0.19, 1.83)	0.357	

*adjusted for age, time from HIV diagnosis, CD4 count nadir, VL<=50 copies/mL at T0, days from the date of 2nd dose, vaccine sequence, and concomitant cancer

&from the adjusted model

In my opinion the revised manuscript still does not clearly address these questions, for the following reasons:

- Though a control group of people without HIV was added, no data from this group was incorporated into any figure or Table, just summarized briefly in the text. This information is very important however, as it appears that cellular (though not humoral) responses were weaker in PLWH compared to controls.

We have now included a new Figure and made the results of the comparison with the general population more prominent in the text section (see lines 300-302, 320-322 and 335-337).

- The authors still do not adjust for vaccine-related variables (including regimen type and dose interval) in their primary multivariable analyses of the relationship between HIV (and CD4 T-cell count in PLWH) and COVID-19 vaccine responses. The authors justify this by stating that they are interested in exploring the effects of vaccine regimen type separately from the effects of HIV/CD4+ T-cell count. But, this is not appropriate for a paper whose major goal is to investigate the effects of HIV (and CD4+ T-cell count) on COVID-19 vaccine responses. Controlling for vaccine regimen is especially relevant given the authors' discovery that vaccine regimen type influenced vaccine responses, and that the three CD4 strata in PLWH differed in terms of their vaccine regimen type (as reported in table 1; p=0.02).

The lack of inclusion of the time elapsed since the last vaccine dose was an oversight as this is a confounder according to our assumptions (see DAG). Our reason for not controlling for type of

vaccination was that this factor was unlikely to have an effect on current CD4 count. However, we have now included both factors in the multivariable model and the association with current CD4 count was further attenuated with all the vaccine response studied. Overall we now conclude that current CD4 count does not appear to affect vaccine response while PCDR participants showed impaired T-cell mediated response as compared to the general population. We have clearly stated these conclusions in the revised Conclusion paragraphs in both the main text and abstract.

- I maintain that it is not appropriate to use a binary classifier (responder/non responder) as a primary analysis when the number of non-responders is essentially zero (as one would expect following three vaccine doses). While the authors can (and should) report rare occurrences of non-response, my point was that the primary statistical analyses should not be based on these. The only multivariable analyses presented in the paper are Tables 2 and 3, which use binary outcome classifiers.

We respectfully disagree with the reviewer on this point. Immunological correlates of protection are currently unclear so that the clinical impact for a difference magnitude of the effect in responses, say observed by CD4 count in our analysis (ranging from 0.4 to 0.6 log₂ difference) is largely unknown. In contrast, there is a large bulk of evidence showing that persons who had no response to the vaccines and with a preexisting immune depressive condition are at increased risk of severe disease so the binary outcome is a lot more informative from a clinical perspective. Having said this, we felt that the mean values of the responses in PLWH by current CD4 count and in HCW were not clearly specified in the Results so we have added more details in the Results section (see lines 368-370, 383-385 and 444-446).

- I acknowledge that the authors do perform multivariable analyses on the quantitative values as secondary analyses, in the form of a "truncated regression analysis". The description of this analysis however, which both reviewers 2 and 3 indicated was unclear, remains unchanged in the revised manuscript (lines 190-193).

We have added additional details on the methodology used. Because of the large number of participants with a response value above the upper limit cut-off of the assay, ANOVA results could be biased. Truncated linear regression adequately controlled for censored data for the outcome variable and the method was used as a sensitivity analysis to provide further evidence in favour or against the null hypothesis of no association. Now it has been further clarified in methods' section (see lines 212-214).

Additional comments:

1. The use of the terminology "third additional dose" is confusing. Reviewer 1 had raised this point, and had suggested the authors use "third dose".

We have modified the wording and use 3rd dose throughout after describing dosage and interval time used in this setting.

2. There are still errors and typos throughout. For example, the Ns reported in the text for vaccine regimen types (lines 232-235) only add up to 212, not the reported total n=216. Moreover, these numbers do not match those in Table 1, which only add up to 214.

There were two participants with a sequence involving not mRNA vaccines (one Astra-Zeneca followed by Moderna and one J&J followed by Moderna) which have been excluded from the regression analysis so the correct number is 216 of whom only 214 were included in the ANOVA analysis. The correct size of the included groups are now corrected over lines 261-263. We apology for the previous discrepancies in text and Tables, reason for which has been now clarified.

Reviewers' Comments:

Reviewer #3:

Remarks to the Author:

The authors have responded to the reviewer concerns and the manuscript is improved.

We thank again the reviewers for their thoughtful insights, which helped to significantly improve the manuscript.